# An extended retinotopic map of mouse cortex

**Jun Zhuang, Lydia Ng, Derric Williams, Matthew Valley, Yang Li, Marina Garrett, Jack Waters***

Allen Institute for Brain Science, Seattle, United States

**Abstract** Visual perception and behavior are mediated by cortical areas that have been distinguished using architectonic and retinotopic criteria. We employed fluorescence imaging and GCaMP6 reporter mice to generate retinotopic maps, revealing additional regions of retinotopic organization that extend into barrel and retrosplenial cortices. Aligning retinotopic maps to architectonic borders, we found a mismatch in border location, indicating that architectonic borders are not aligned with the retinotopic transition at the vertical meridian. We also assessed the representation of visual space within each region, finding that four visual areas bordering V1 (LM, P, PM and RL) display complementary representations, with overlap primarily at the central hemifield. Our results extend our understanding of the organization of mouse cortex to include up to 16 distinct retinotopically organized regions.

*For correspondence: jackw@alleninstitute.org

**Competing interests:** The authors declare that no competing interests exist.

## Introduction

Mammalian visual cortex consists of a series of interconnected areas, which correspond to topographically-organized collections of neurons with similar functional properties and patterns of connectivity (*Van Essen, 2003*). In mouse neocortex, 12 visual areas have been identified, each corresponding to a retinotopically-organized map of the visual field (*Dräger, 1975*; *Wagor et al., 1980*; *Olavarria et al., 1982*; *Olavarria and Montero, 1989*; *Schuett et al., 2002*; *Kalatsky and Stryker, 2003*; *Wang and Burkhalter, 2007*; *Marshel et al., 2011*; *Wang and Burkhalter, 2013*; *Garrett et al., 2014*; *Tohmi et al., 2014*).

The borders between visual areas have been located using architectonic and retinotopic criteria. Architectonic borders are associated with changes in staining (chemoarchitectonics), cell density (cytoarchitectonics), or myelination patterns (myeloarchitechtonics). Retinotopic borders are associated with a change in chirality of the retinotopic map and have been identified anatomically by labeling projections between visual areas, and functionally by recording the responses of neurons with electrodes or imaging techniques. The architectonic and retinotopic borders of visual areas in the mouse are generally thought to be co-aligned, but the shapes and locations of visual areas can differ across studies.

We generated retinotopic maps from GCaMP6 transgenic mice (*Chen et al., 2013*; *Madisen et al., 2015*) and compared functional retinotopic, anatomical retinotopic, chemoarchitectonic, cytoarchitectonic and myeloarchitechtonic maps. Our results revealed new regions of retinotopic organization in mouse neocortex, some within and some outside the accepted boundaries of mouse visual cortex, and a mismatch between retinotopic and architectonic maps that may help reconcile the differences between visual area maps generated by these complementary approaches.

**eLife digest** Our eyes send information about the world around us to a region on the surface of the brain called the visual cortex, which is made up of a series of interconnected areas. Researchers have studied the anatomy and activity of these areas to generate maps that show how these areas are arranged. For example, architectonic borders are drawn where there are abrupt changes in the density of cells, or the degree to which they are stained by certain chemicals. Maps based on architectonics and those based on brain activity are thought to exhibit matching borders between visual areas.

Zhuang et al. used animaging approach to produce detailed maps of the visual cortex of mice. The approach uses a fluorescent protein called GCaMP6 to indicate levels of activity in the brain while the mice were exposed to visual cues. Furthermore, Zhuang et al. added a second step to this approach to reveal the architectonic borders of the areas in the visual cortex. This made it possible to compare the locations of activity-based and anatomical borders in a single mouse.

Zhuang et al. found that maps of the visual cortex based on architectonics do not completely match those based on activity. These findings help reconcile the differences between maps of mouse visual cortex produced by other studies. It is not clear whether a similar mismatch in architectonic and activity-based border locations exists in other animals, such as primates.

## Results

### GCaMP6 expression patterns and fluorescence signals

We mapped visual areas in three GCaMP6 reporter lines: Ai95(RGL-GCaMP6f), Ai96(RGL-GCaMP6s) and Ai93(TITL-GCaMP6f). Each reporter line was crossed with the Emx1-IRES-Cre line, driving GCaMP6 expression in pyramidal neurons in all layers of neocortex (*Figure 1—figure supplement 1*). We imaged through a 5 mm diameter glass window implanted over visual areas (*Figure 1A,B*). The time-averaged fluorescence, measured in primary visual cortex in the absence of visual stimuli, was greater in all three reporter lines than in wild-type mice (*Figure 1H*; fluorescence normalized to wild-type, Emx1-Ai95 10.3 ± 1.3, Emx1-Ai96 4.7 ± 0.5, Emx1-Ai93 49.7 ± 5.9).

To measure response amplitudes and kinetics, we presented mice with a brief visual stimulus consisting of a white circle on a black background (diameter 20 degrees; center location 60° azimuth, 0° altitude; 50 ms duration). In wild-type mice, brief stimuli evoked a decrease in fractional fluorescence of −2.3 ± 0.3% △F/F (four mice, *Figure 1F*). In all three mouse lines we observed transient activation of visual areas (*Figure 1C-E*) that was larger and faster than autofluorescence signals in wild-type mice (*Figure 1F,G,I,J*), followed by a decline in fluorescence that may result from vasodilation (*Pisauro et al., 2013*). In Emx1-Ai93 mice, brief stimuli evoked a peak fractional fluorescence change of 10 ± 2.4% △F/F (range 5.7–15.0% △F/F, 4 mice, *Figure 1F*) and a mean peak change in absolute GCaMP6 fluorescence that was 216 times the peak fluorescence change observed in wild-type mice (*Figure 1G*). All GCaMP6 mice displayed kinetics that were faster than the autofluorescence signal in wild-type mice (latency to peak from stimulus onset of 252 ± 49 ms, 6 Emx1-Ai95mice; 414 ± 60 ms, 4 Emx1-Ai96 mice; 219 ± 60 ms, 4 Emx1-Ai93 mice; 1.37 ± 0.3 s, four wild-type mice; mean 10–90% rise and decay times 71 and 536 ms in Emx1-Ai95 mice, 225 and 1151 ms in Emx1-Ai96 mice, 69 and 752 ms in Emx1-Ai93 mice). In summary, all three Emx1-GCaMP6 mouse lines displayed brighter fluorescence and faster fluorescence changes than wild-type mice with the result that the changes in fluorescence were dominated by GCaMP6.

### Retinotopic maps from GCaMP6 fluorescence reveal additional patches of retinotopic organization

To generate retinotopic maps, we employed a spherically-corrected checkerboard visual stimulus drifting across the visual field at 0.043–0.048 Hz (*Figure 2A*) (*Kalatsky and Stryker, 2003*; *Marshel et al., 2011*) and mapped retinotopy in Emx1-GCaMP6 mice that were head-restrained and free to run on a rotating disk. The mean fluorescence change was greatest in visual cortex (*Figure 2B*), where the signal-to-noise ratio was sufficient to identify visually-evoked changes in

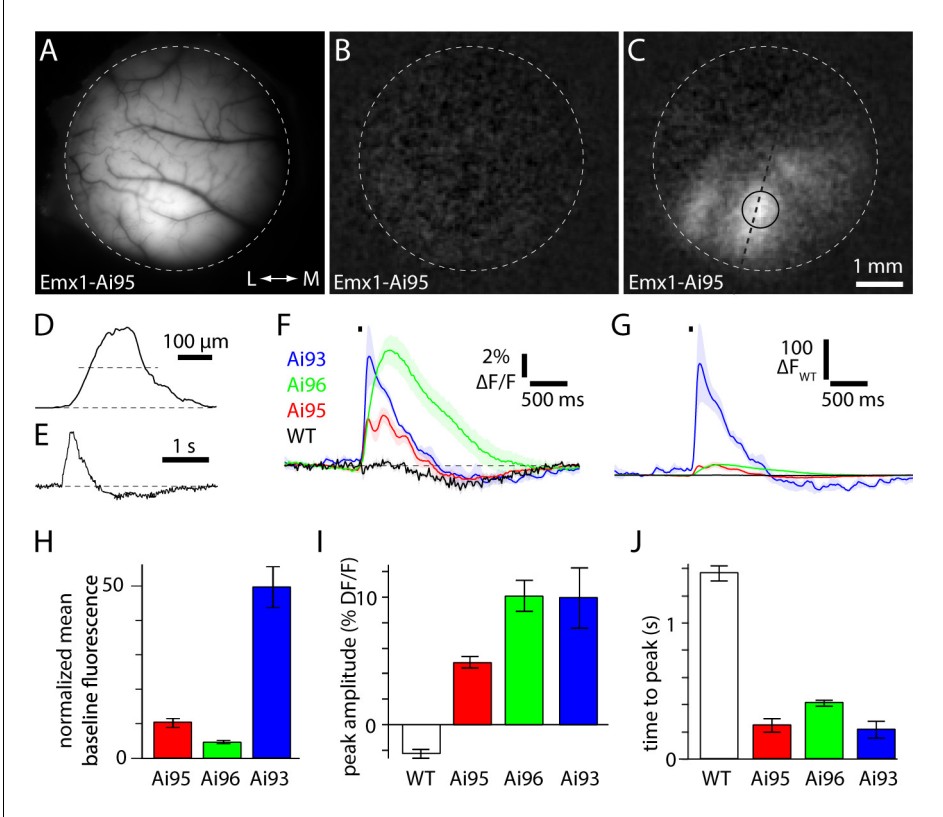

**Figure 1.** Amplitude and kinetics of fluorescence transients to brief visual stimuli. (A) Fluorescence image focused on the cortical surface of an Emx1-Ai95 mouse. Approximate edge of 5 mm diameter craniotomy is marked with a dashed line. (B and C) Baseline (B) and peak (C) change in fluorescence for the field of view shown in A. In C, dashed line and circle indicate the regions used to extract values for D and E, respectively. (D) Spatial extent of the fluorescence change along line marked in panel C. Width of response at half height (dashed line) is 190 µm. (E) Fluorescence time course for a single trial, from the region marked in panel C. D and E are in arbitrary fluorescence units. (F) Mean ± SEM fractional fluorescence changes for each mouse line. Four wild-type mice (black), 6 Emx1-Ai95 mice (red), 4 Emx1-Ai96 mice (green), 4 Emx1-Ai93 mice (blue). Stimulus indicated with black bar. (G) Mean change in fluorescence ($\triangle F$) for each mouse line, normalized to the peak amplitude of the fluorescence change in wild-type mice ($\triangle F_{WT}$). Shaded areas denote ± SEM. (H) Time-averaged baseline fluorescence, normalized to wild-type mice. Bears denote mean ± SEM from four wild-type mice, 6 Emx1-Ai95 mice, 4 Emx1-Ai96 mice, 4 Emx1-Ai93 mice. (I) Peak fractional fluorescence change. (J) Time to peak fluorescence, measured from the onset of the stimulus.

The following figure supplement is available for figure 1:

**Figure supplement 1.** Expression of GCaMP6 in Emx1-Ai95, Emx1-Ai96 and Emx1-Ai93 mice under the Emx1-IRES-Cre driver.

fluorescence in individual trials (*Figure 2C*). To generate retinotopic maps, we averaged 10–40 presentations of the stimulus in each of the four cardinal directions, thereby reducing the potential effects of ongoing activity (*Figure 2D*; *Video 1*). We generated azimuth and altitude position maps (*Kalatsky and Stryker, 2003*) that included retinotopic locations across the visual field from approximately −20 to +30 degrees in altitude and −10 to +90 degrees in azimuth (*Figure 2E,F*). As described previously (*Sereno et al., 1994*; *1995*; *Garrett et al., 2014*), phase maps were consolidated into a visual field sign map (*Figure 2G*).

The borders between field sign patches (*Figure 2H*) were identified using a numerical routine (*Figure 2—figure supplement 1*). Briefly, the sign map was lightly filtered and thresholded to create an initial parcellation of cortex into patches. The threshold was tuned manually, with little change in

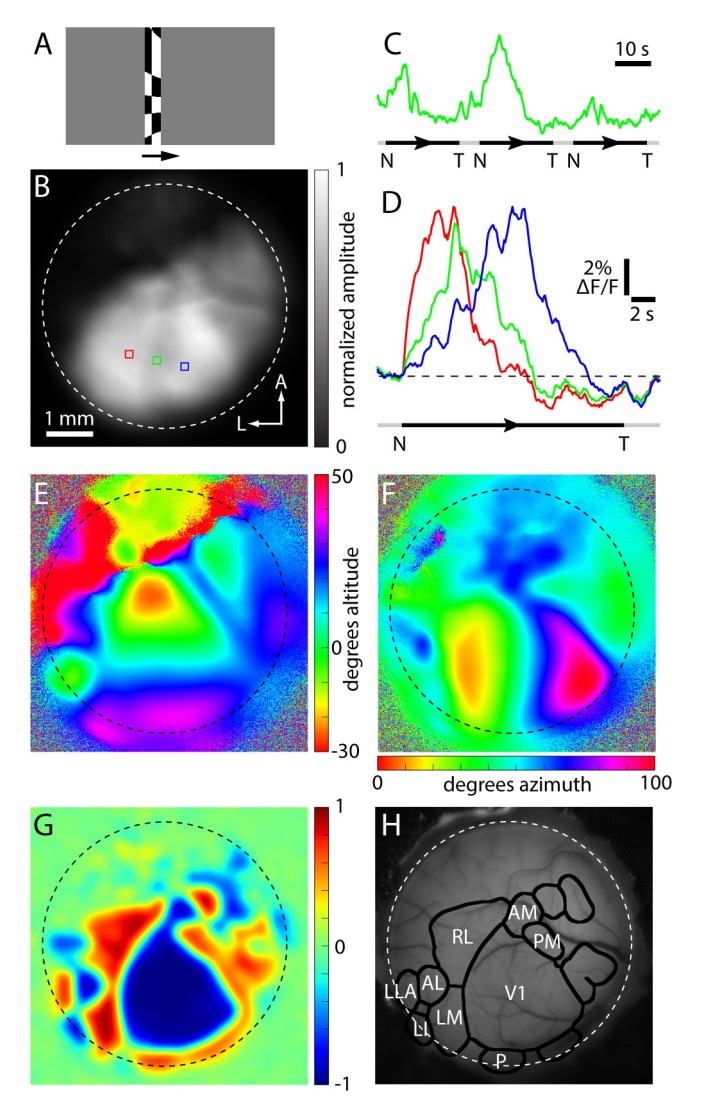

**Figure 2.** Example of a GCaMP6 fluorescence-based retinotopic map. Example of a GCaMP6 fluorescence-based retinotopic mapping data set from an awake mouse, generated with the drifting checkerboard stimulus on a grey background. (**A**) A single image from the visual stimulus movie used to map in the nasal to temporal (azimuth) direction in an Emx1-Ai96 mouse. The checkerboard pattern was swept from left to right (arrow) on a grey background. (**B**) Greyscale image illustrating the amplitude of the fluorescence change at 0.043 Hz during azimuth mapping, normalized to the maximum amplitude in the image. Dashed white circle approximates the border of the cranial window. (**C**) Spatial average fluorescence during nasal (N) to temporal (T) mapping from the region of interest outlined in green in panel **B**. The timing of stimulus presentation is indicated below, where the black segments indicate the center position of the checkerboard bar from −14 to 132 degrees (azimuth) and the grey segments indicate that no stimulus was on the monitor. (**D**) Fractional fluorescence changes from the three regions marked in panel **B**. Each trace is the average of 10 presentations of the stimulus. (**E** and **F**) Altitude and azimuth maps for the same cranial window. (**G**) Field sign map derived from the altitude and azimuth maps. (**H**) The result of automated border identification, drawn on a brightfield image of the brain surface over visual areas. Named visual areas were identified manually, based on published maps of visual areas.

The following figure supplements are available for figure 2:

**Figure supplement 1.** Schematic summary of border identification routine.

**Figure supplement 2.** Effects of sign map threshold on patch size, shape and visual coverage.

*Figure 2 continued on next page*

*Figure 2 continued*

**Figure supplement 3.** Eye movements and pupil area during retinotopic mapping.

**Figure supplement 4.** Comparison of maps with and without eye movements.

**Figure supplement 5.** Comparison of fluorescence changes and V1 coverage with stimuli on black and grey backgrounds.

the incidence, shape, border locations and area of most patches over the range of threshold values employed (0.2–0.4, mean ± SEM of 0.32 ± 0.004, 14 mice; *Figure 2—figure supplement 2*). Patches were split and merged to ensure that neighboring patches with the same sign had redundant representation of visual space of ≤10% (see Materials and methods).

If sufficiently large and stereotyped, changes in pupil location or size might confound generation of accurate retinotopic maps in awake mice. During mapping, we observed abrupt eye movements of up to ~5–10 degrees vertically and up to ~15–20 degrees horizontally (*Figure 2—figure supplement 3C*). Movements occurred during all phases of the stimulus and mean displacement was <~1 degree (*Figure 2—figure supplement 3D*), leading us to conclude that movement-related effects were eliminated by averaging in our experiments (*Figure 2—figure supplement 4*). To minimize systematic changes in pupil size, we employed a stimulus with a constant mean luminance, delivering the moving checkerboard pattern on a grey background with 50% of the maximum luminance (*Figure 2—figure supplement 5*, *Videos 2,3*). Hence, in our experiments maps were largely unaffected by eye movements or changes in pupil size.

Twelve discrete areas have been identified in mouse visual cortex: primary visual cortex (V1), lateromedial area (LM), rostrolateral area (RL), anterior area (A), anteromedial area (AM), posteromedial area (PM), medial area (M), posterior area (P), postrhinal area (POR), laterointermediate area (LI), anterolateral area (AL), and laterolateral anterior area (LLA) (*Dräger, 1975*; *Wagor et al., 1980*; *Olavarria et al., 1982*; *Olavarria and Montero, 1989*; *Wang and Burkhalter, 2007*; *Garrett et al., 2014*). Each area contains one retinotopic map and therefore appears in field sign maps as a single, distinct, positive or negative field sign patch (*Sereno et al., 1994*, *1995*). Across mice these patches are arranged in a stereotyped configuration with consistent positioning relative to each other. To identify regions in our maps objectively, we used the automated image analysis approach described by *Garrett et al. (2014)*, which identifies field sign patches that each contain one and only one map of retinotopic space.

Most field sign patches were visible after only 15 min of imaging (*Figure 3A*; 10 sweeps of the stimulus in each direction) and maps were stable across imaging sessions (*Figure 3—figure supplement 1*). The ability to quickly generate maps that are comparable to those resulting from 1–2 hr of red-wavelength reflectance-based imaging under anesthesia (*Garrett et al., 2014*) is a practical advantage of GCaMP6 fluorescence-based mapping. Typically, we further reduced the effects on ongoing activity by presenting 20–40 sweeps of the stimulus in each direction (imaging for 30–60 min). Like previous authors, we found that maps displayed consistent structure across mice (*Figure 3A*). The arrangement of areas was broadly consistent with published maps of the mouse visual system (e.g. *Wang and Burkhalter, 2007*; *Marshel et al., 2011*; *Garrett et al., 2014*), but differed in several respects. Key differences include additional visual field sign patches, particularly medial to AM and PM (*Figure 3A*). The narrow strip of

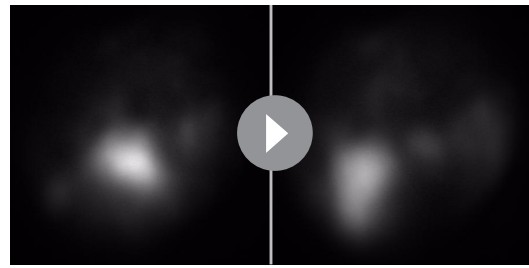

**Video 1.** Example fluorescence movies from retinotopic mapping experiment.    Example of fluorescence changes during retinotopic mapping. Left: response to checkerboard travelling from the lower to the upper visual field. Right response to checkerboard travelling from the nasal to temporal visual field. Each movie is the mean change in fluorescence (△F) of 40 trials.

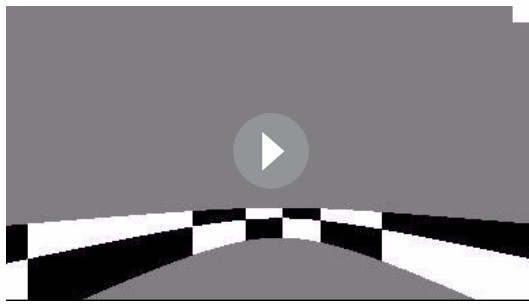

**Video 2.** Lower-to-upper visual field visual stimulus. Lower to upper visual field moving checkerboard stimulus used for retinotopic mapping.

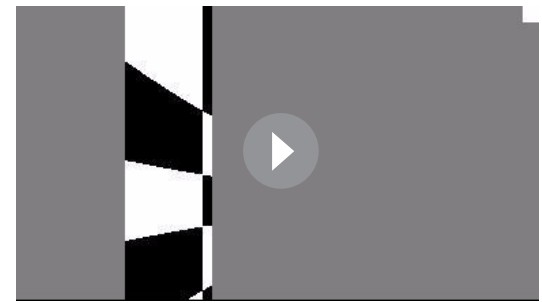

**Video 3.** Nasal-to-temporal visual stimulus. Nasal-to-temporal moving checkerboard stimulus used for retinotopic mapping.

tissue between AM/PM and retrosplenial cortex is generally termed MM or V2MM and is considered part of visual cortex (*Wang and Burkhalter, 2007*; *Franklin and Paxinos, 2007*), but its retinotopic structure remains uncharacterized. GCaMP6 fluorescence-based maps revealed consistent structure in MM, with a positive field sign patch medial to AM (MMA) and a negative field sign patch medial to PM (MMP). Furthermore, there were one or more positive field sign patches medial to MMA and MMP, likely in retrosplenial cortex. The most consistent was a positive field sign region which our automated segmentation routine generally failed to separate from PM. Finally, in lateral visual cortex, we observed a negative field sign patch anterior and lateral to RL, which we termed RLL.

MMA, MMP and RLL appear in the mean field sign map, which summarizes the locations of field sign patches that mapped consistently across 14 Emx1-Ai96 and Emx1-Ai93 mice (*Figure 3C*). Across these 14 mice, the maximum numbers of patches in an individual map were nine for positive and seven for negative field sign patches. Maps from Emx1-Ai96 and Emx1-Ai93 mice were similar (*Figure 3B*), with 11.75 field sign patches in Emx1-Ai96 (range 11–14, 4 Emx1-Ai96 mice) and 12 field sign patches in Emx1-Ai93 mice (range 9–14, 10 Emx1-Ai93 mice).

Our results indicate that there is retinotopic organization in regions of the mouse cortex in which retinotopy had not been reported. In our mouse lines, GCaMP6 is present in neuronal somata and dendrites (*Figure 1—figure supplement 1*) and presumably in axons. Accordingly, some of these new field sign patches may result from retinotopically-organized projections originating in surrounding visual areas. Since our results do not indicate whether somata in these regions are retinotopically-organized, we refer to these regions of extended retinotopic organization as 'patches' rather than visual 'areas'.

We performed further calculations to characterize and visualize mouse-to-mouse variability and test the consistency of each field sign patch. To visualize mouse-to-mouse variability, we calculated a variance map in which the intensity of each pixel represents the variance of the field sign (*Figure 3D*). Variance was lower towards the centers of field sign patches and higher towards the borders, but there was little difference in the mean variance per unit area between areas (not shown), except for V1, which displayed low variance per unit area.

To quantify mouse-to-mouse consistency in area identification, we calculated the incidence of each field sign patch (*Figure 3E*). 10 field sign patches (V1, LM, LI, AL, RL, AM, PM, MMA, MMP, P) occurred in >85% of Emx1-Ai93 and Emx1-Ai96 mice (≥12 of 14 mice). LI and LLA were frequently along the lateral edge of the cranial window, which may account for the lower incidence of LLA in our maps (10 of 13 mice). Similarly, M was near the postero-medial edge of the window and was observed in 5 of 9 mice (in 4 of 7 Emx1-Ai93 and 1 of 4 Emx1-Ai96 mice). POR was absent from our maps, probably because it was invariably outside the cranial window.

Of the areas that were within the cranial window, the anterior patches A and RLL occurred least consistently. Area A is a putative negative field sign region medial to RL (*Wang and Burkhalter, 2007*) which maps inconsistently with red-wavelength reflectance and autofluorescence imaging in anesthetized mice (*Marshel et al., 2011*; *Garrett et al., 2014*; *Tohmi et al., 2014*). We observed a negative field sign patch in this location in 1 of 14 mice and a positive field sign patch (which

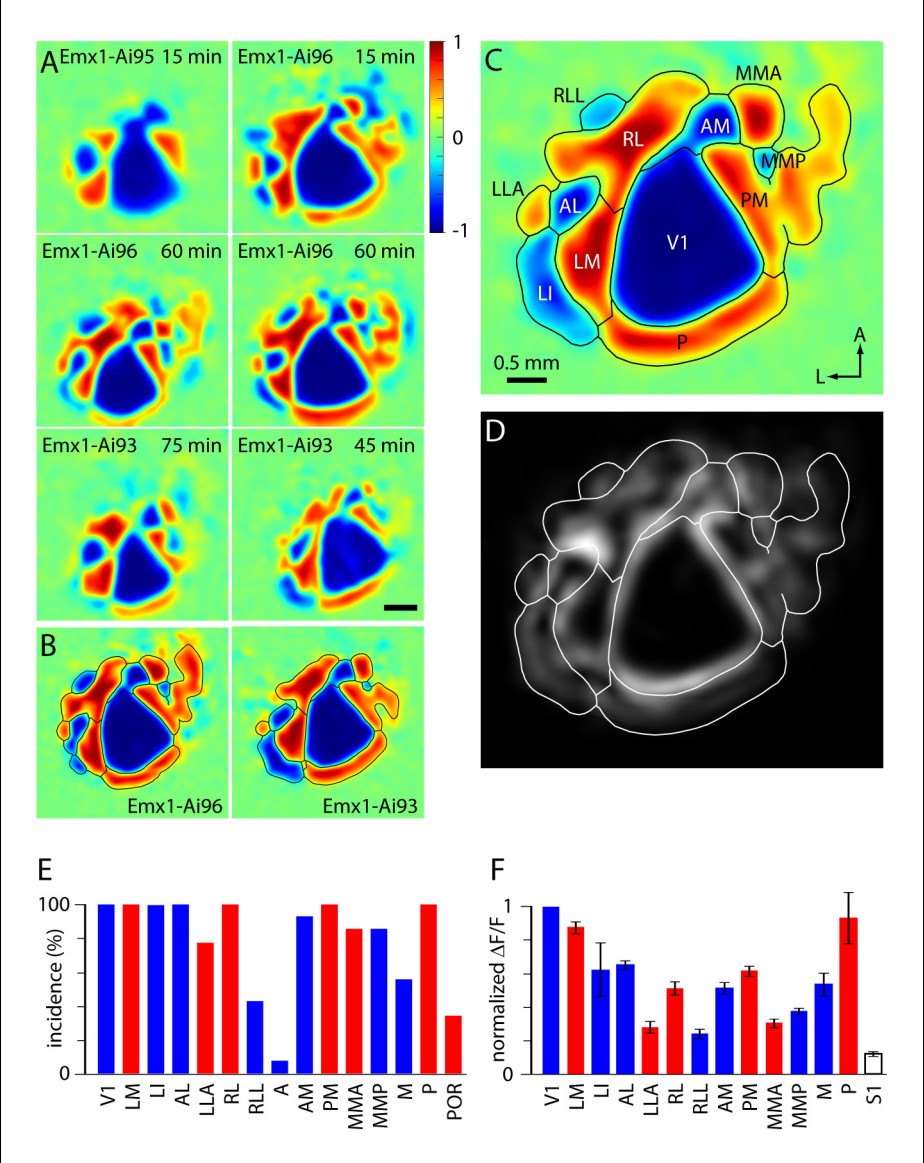

**Figure 3.** Retinotopic organization of mouse visual cortex. (**A**) Field sign maps from six mice, illustrating differences between mouse lines and individual mice. The mouse line and duration of imaging are indicated on each map. Scale bar 0.5 mm. (**B**) Mean field sign maps for 4 Emx1-Ai96 and 10 Emx1-Ai93 mice, from 30–75 min of imaging. (**C**) Mean of the Emx1-Ai96 and Emx1-Ai93 field sign maps in panel B, with borders and area labels. (**D**) Map of variance of the visual field sign. Variance was calculated from visual field sign maps from 14 mice, after alignment as described for calculation of the mean field sign map. Whiter areas denote higher variance. Area borders are overlaid in white. (**E**) The probability of mapping different visual areas with GCaMP6 fluorescence in awake mice. Blue and red bars denote areas with negative and positive field signs, respectively. Results were derived from 14 mice (4 Emx1-Ai96, 10 Emx1-Ai93). Mouse numbers: V1 14/14, LM 14/14, LI 10/10, AL 14/14, LLA 10/13, RL 14/14, RLL 6/14, AM 13/14, PM 14/14, MMA 14/14, MMP 14/14, M 5/9, P 14/14, POR 1/3, where, for each area, the denominator indicates the number of mice in which the area was visible within the cranial window, determined manually. (**F**) Mean ± SEM fluorescence change for each visual area, derived from the △F/F spectral power. For each map power was normalized to that in V1. S1 region was drawn manually towards the anterior extent of the cranial window. Mouse numbers: LM 14, LI 10, AL 14, LLA 10, RL 14, RLL 5, AM 12, PM 14, MMA 12, MMP 12, M 5, P 14, S1 14; from 14 mice (4 Emx1-Ai96, 10 Emx1-Ai93).

The following figure supplement is available for figure 3:

**Figure supplement 1.** Map stability.

segmented separately from RL) in 3 of 14 mice (2 of 10 Emx1-Ai93 mice and 1 of 4 Emx1-Ai96 mice). The presence of area A in GCaMP6 fluorescence maps from only 1 of 14 mice probably reflects the difficulty of mapping this area, in which projections from V1 are diffuse and their topographic organization appears weaker than that of many other areas (*Wang and Burkhalter, 2007*). RLL was observed in 60% of mice (3 of 10 Emx1-Ai93 and 3 of 4 Emx1-Ai96 mice). In summary, of the three new field sign patches, MMA and MMP appeared consistently across mice, whereas RLL was less consistent.

We further tested the consistency of location of each patch in the mean sign map using k-means cluster analysis, as described previously (*Garrett et al., 2014*). Cluster metric ($C_k$) values were LM 0.23; LI 0.42; AL 0.36; LLA 1.17; RL 0.42; RLL 1.29; AM 0.77; PM 0.63; MMA 0.69; MMP 1.0; P 0.39. (Maps were aligned to the centroid of V1.) Of the three new patches, MMA and MMP were each associated with a tightly-packed cluster of patch centroids that differed from a shuffled distribution. In contrast, and consistent with its relatively low incidence, RLL was associated with a non-significant cluster ($C_k = 1.29$).

To summarize our results for the three new field sign patches, MMA and MMP mapped consistently: they appear on the mean field sign map, were present in almost every mouse and their locations relative to other field sign patches were sufficiently consistent that they were associated with a significant k-means cluster. In contrast, RLL was consistent enough to appear on the mean field sign map, but was present in only 6 of 14 mice and was associated with a non-significant k-means cluster.

Our ability to map additional regions of retinotopic organization likely results from the superior signal-to-noise ratio of GCaMP6 fluorescence-based maps relative to intrinsic signals. Consistent with this hypothesis, the amplitudes of fluorescence changes were smaller in MMA, MMP, RLL and LLA than for other visual field sign patches (*Figure 3F*). Of these four patches, LLA was first mapped recently with red reflectance-based imaging and extensive averaging (*Garrett et al., 2014*) and the remaining three have now been revealed with GCaMP6 fluorescence mapping. RLL exhibits the smallest mean change in fluorescence of any of the visual regions within our maps (*Figure 3F*) and this weak activation, and resulting low signal-to-noise ratio, likely accounts for the inconsistency of RLL in our maps (*Figure 3E*).

## Relationship between retinotopic and architectonic borders

The new field sign patches identified in our GCaMP6 fluorescence-based maps extend far beyond the borders of V1, in some cases by >2 mm. It is likely that some of these newly-mapped patches extend into architectonically-defined cortical areas surrounding visual cortex. For example, AM and PM are separated from retrosplenial cortex by V2MM. This narrow strip of tissue would appear large enough to accommodate MMA and MMP, but not the long medial extension of PM, which is likely, therefore, to be within retrosplenial cortex. Similarly, the architectonically-defined borders of V1 and barrel cortex are separated by a thin band of tissue, raising the possibility that RLL, and perhaps RL, might extend into barrel cortex.

To compare retinotopic and chemoarchitectonic borders, after retinotopic mapping we processed tissue for cytochrome C oxidase staining. We labeled surface vasculature with a fluorescent dye by transcardial perfusion immediately before fixation, used the vasculature to align images acquired in vivo (*Figure 4A*; *Figure 4—figure supplement 1*), after fixation (whole mount, *Figure 4B*), after flattening (*Figure 4C*) and after sectioning parallel to the cortical surface and staining (*Figure 4D*) and compared retinotopic and chemoarchitectonic borders from 4 Emx1-Ai96 mice (*Figure 4G–J*). The results confirm that the retinotopic map extends into primary somatosensory cortex and retrosplenial cortex (*Figure 4G–I*), with RLL mapping to the posterior whiskers (posterior barrels) of barrel columns B and C.

Another prominent feature of the comparison between cytochrome oxidase-based chemoarchitectonic and GCaMP6-based functional maps is a mismatch in the location of the lateral border of V1. The lateral border of chemoarchitectonically-defined V1 runs through functionally-defined areas RL and LM. The distance between the chemoarchitectonically-defined lateral border of V1 and that defined by functional retinotopy, measured along the LM/RL border, was 312 ± 88 µm (4 Emx1-Ai96 mice).

One possible explanation for the mismatch is that the cytochrome oxidase-rich region of visual cortex extends beyond the reversal in retinotopy that defines the functional border of V1. Alternatively, the apparent mismatch in border locations might be an experimental artifact arising from

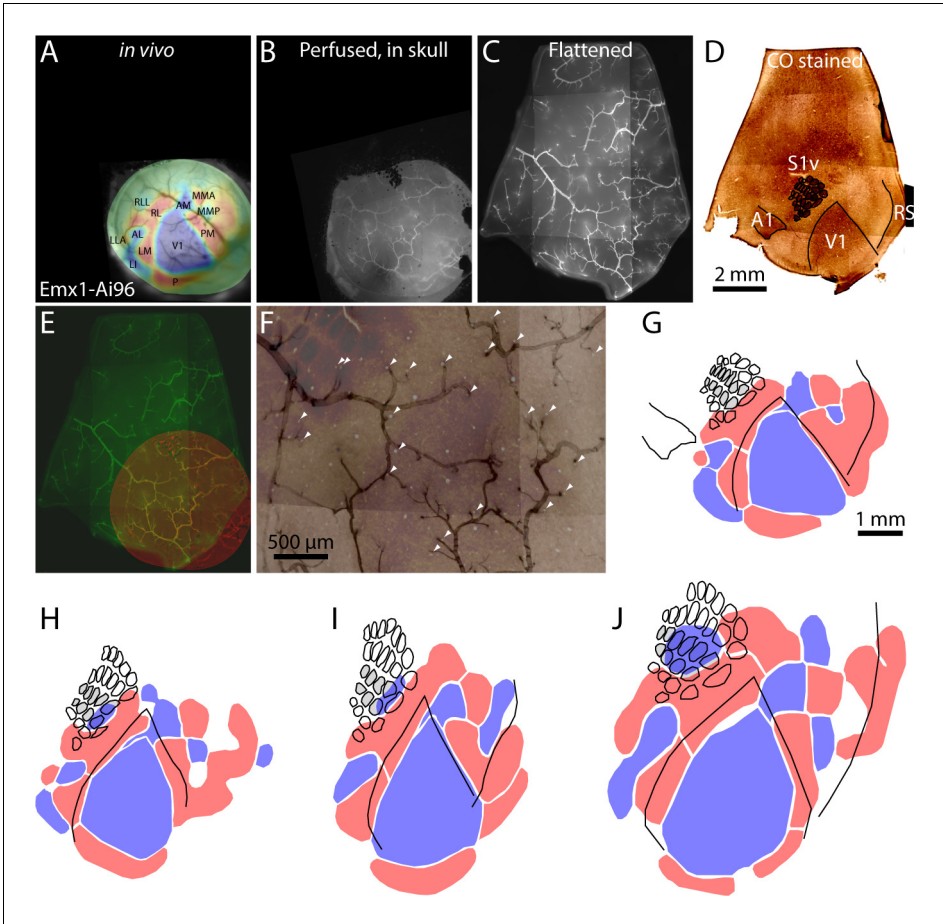

**Figure 4.** Registration of functional retinotopic maps to chemoarchitectonic borders. (A–D) Images from key stages in the processing of tissue from an Emx1-Ai96 mouse, each aligned to the cytochrome C oxidase (CO) image. (A) Brightfield image of surface vasculature with overlaid field sign map. (B) Fluorescence image of whole-mount brain, after perfusion, in which a subset of the surface vasculature is labeled with DyLight 649-lectin conjugate. (C) fluorescence image of the flattened cortex. D: brightfield image of a section through layer four after CO staining. (E) Overlaid fluorescence images of surface vasculature in whole-mount (red, panel **B**) and after flattening (green, panel **C**). (F) Overlaid images of the surface vasculature and CO staining in posterior barrel cortex and anterior V1. The contrast of the vasculature image is inverted for clarity. Arrowheads indicate small, circular regions that do not stain for CO and likely result from transverse cuts through ascending/descending vessels. Note the alignment of these putative vessels with likely locations of ascending/descending vessels in the fluorescence image of surface vasculature. (G) Field sign map (panel **A**) aligned to chemoarchitectonic borders from the CO image (panel **D**). Borders of primary visual cortex, auditory cortex, and of barrels in primary somatosensory cortex) were drawn manually. Barrels in putative columns B and C are shaded grey. (H–J) Alignment of functional retinotopic maps and chemoarchitectonic borders for three additional Emx1-Ai96 mice.

The following figure supplement is available for figure 4:

**Figure supplement 1.** Vessels common to images from live and fixed tissue.

misalignment of the maps during fixation and subsequent histological processing, but misalignment on this scale is unlikely given our fluorescent marker-based registration process and direct alignment of images from live and fixed tissue.

To further clarify the alignment of functional and architectonic boundaries, we identified cytoarchitectonic borders in Rorb-Ai93 mice. Rorb drives expression preferentially in layer four and upper layer five pyramidal neurons, leading to stronger labeling of primary sensory areas than of surrounding cortex (http://connectivity.brain-map.org/transgenic/search?page_num=0&page_size=29&no_

paging=false&search_type=line-name&search_term=Rorb-IRES2-Cre). Resting GCaMP6 fluorescence was greater in primary sensory areas than in surrounding regions (*Figure 5A*), enabling cytoarchitectonic borders to be identified by thresholding GCaMP6 fluorescence images (*Figure 5B*). Consequently, GCaMP6 fluorescence-based retinotopic maps were inherently aligned to cytoarchitectonic borders in Rorb-Ai93 mice, eliminating the potential for misalignment resulting from tissue processing. Comparison of cytoarchitectonic and functional retinotopic borders again indicated a mismatch, with the lateral border of cytoarchitectonically-defined V1 being lateral of the functionally-defined border and running through functionally-defined areas RL and LM (*Figure 5C, D*). The distance between the cytoarchitectonically-defined lateral border of V1 and that defined by functional retinotopy, measured along the LM/RL border, was 120 ± 38 μm (10 Rorb-Ai93 mice).

A similar change in resting GCaMP6 fluorescence was observed at the borders of primary sensory areas in Emx1-Ai93 mice (*Figure 1—figure supplement 1F,G*). The change in fluorescence was less pronounced than in Rorb-Ai93 mice, but borders were visible after averaging images of the resting GCaMP6 fluorescence from 10 Emx1-Ai93 mice (*Figure 5E*). Here again, the lateral border of cytoarchitectonically-defined V1 was lateral of the functionally-defined border of V1 (*Figure 5F*), by 236 μm (measured along the LM/RL border).

Three methods have been particularly influential in the generation of established maps of the mouse visual cortex: architectonics, using various stains, including cytochrome oxidase; functional retinotopy, primarily measured with electrodes; and projection-based retinotopy, in which axonal projections between visual areas were used to identify locations with matching retinotopy. Having measured the mismatch between borders based on architectonics and on functional retinotopy, we next sought to determine whether borders based on projection-based retinotopy more closely match functional retinotopic borders or architectonic borders, and particularly to determine whether there is a mismatch between the lateral border of V1 from projection-based retinotopy and that based on architectonics.

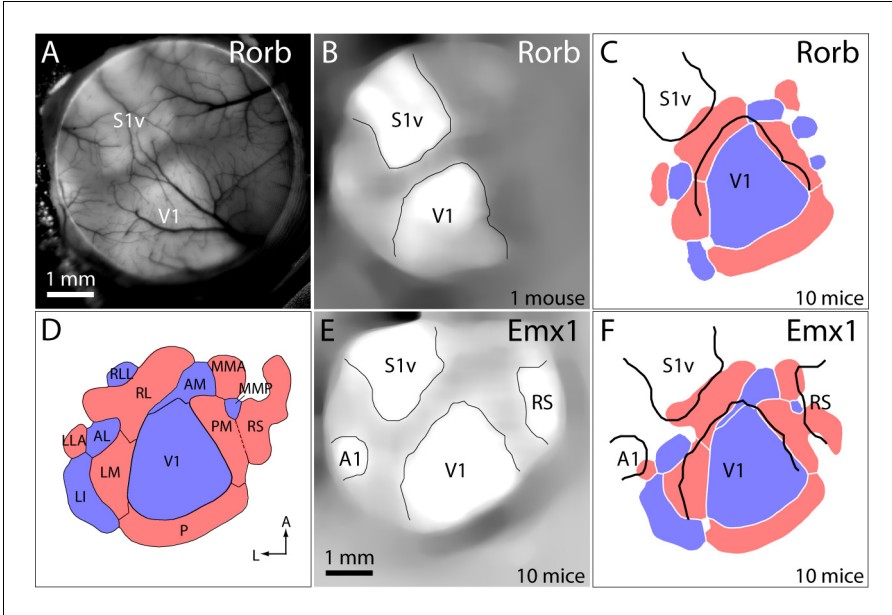

**Figure 5.** Registration of functional retinotopic maps to cytoarchitectonic borders. (**A**) GCaMP6 fluorescence image from a Rorb-Ai93 mouse. Primary sensory areas are marked: S1v barrel cortex, V1 primary visual cortex. (**B**) Image in panel A after filtering and semi-automated identification of major cytoarchitectonic borders. (**C**) Mean retinotopic map/cytoarchitectonic border registration for 10 Rorb-Ai93 mice. Retinotopic maps and cytoarchitectonic borders were pooled across mice as described in the Materials and methods. Cytoarchitectonic borders are shown in black. (**D**) Mean sign map with patch notation, from *Figure 3C*. (**E**) Mean fluorescence image from 10 Emx1-Ai93 mice, after filtering, alignment and semi-automated identification of the borders of primary sensory areas and retrosplenial cortex (RS). (**F**) Mean map/border registration for 10 Emx1-Ai93 mice.

To investigate the alignment of architectonic and projection-based retinotopic borders, we used data from the Allen Mouse Brain Connectivity Atlas (*Oh et al., 2014*; http://connectivity.brain-map.org/). We selected projection data from 99 mice, each with a single injection of anterograde fluorescent tracer into V1. These injections, and the resulting projection maps, were registered to a three-dimensional reference atlas of the mouse brain built from serial section image data sets from 1675 mice. The reference atlas includes tissue autofluorescence images and in the top projection, the major cytoarchitectonic areas (including primary visual cortex, whisker and digit barrels in primary sensory cortex, primary auditory cortex and retrosplenial cortex) are readily visible (*Figure 6A*) due to the enhanced autofluorescence of regions with increased myelination.

For each mouse with an injection into V1, we calculated the density of projections from V1 and combined the results across mice to derive the location within V1 with which each voxel within the brain was most strongly connected (*Figure 6—figure supplement 1*). After projecting the maximum connectivity in superficial cortex to the brain surface, we plot two pseudo-colored top-views in which color indicates the location in V1 to which each pixel in the image was most strongly connected (*Figure 6B,C*). Due to the retinotopic organization of V1, these two plots are analogous to maps of altitude and azimuth retinotopy (*Figure 3E,F*). From these plots we generated the projection-based equivalent of a field sign map, a 'projection sign map' (*Figure 6D*), used our numerical routine to derive area borders (*Figure 6E*) and overlaid these borders onto the autofluorescence top projection from the reference atlas (*Figure 6E*). Due to the registration of all images in the Connectivity Atlas to the 3D reference atlas, the projection sign map and area borders are inherently aligned.

Again, the results indicate that retinotopy extends into primary somatosensory and retrosplenial cortices, with RLL mapping to the posterior whiskers in posterior barrel cortex. LLA also appears to extend into auditory cortex. Regarding the relative positions of the borders of V1, the myeloarchitectonically-defined lateral border of V1 (here visible as a transition between relatively bright autofluorescence in V1 and dimmer autofluorescence more laterally) was lateral to the lateral border of V1 defined by projection-based retinotopy (*Figure 6E,F*). The distance between the myeloarchitectonically-defined lateral border of V1 and that defined by projection-based retinotopy, measured along the LM/RL border, was 180 µm.

The above comparisons of retinotopic maps with architectonic borders using three different methods lead us to two main conclusions. Firstly, retinotopic organization extends into retrosplenial and primary somatosensory cortices in the mouse. RLL is entirely within barrel cortex. The retinotopic map observed in retrosplenial cortex was often continuous with the map in PM, and was not segmented into two distinct regions by our algorithm. The lateral architectonic boundary of retrosplenial cortex extends from the posterior extent of MMP and separates the patch identified as PM into two pieces: a lateral portion that runs parallel to the medial border of V1 and is similar to PM as described previously, and a medial extension within retrosplenial cortex. In subsequent figures we separate these two portions of PM along the approximate lateral boundary of retrosplenial cortex. Secondly, there is a mismatch between the borders of V1 as defined by architectonic markers and by retinotopy, with the architectonically-defined border of V1 lateral to the retinotopic border of V1 by up to ~300 µm (four measurements: 312, 120, 236 and 180 µm; mean 212 µm).

## Single-cell retinotopy along the V1-LM border

With what precision can we locate borders with widefield imaging, which presumably reports the mean retinotopy of many neurons at each location? To answer this question, we compared widefield retinotopic maps with the receptive fields of layer 2/3 pyramidal neurons, measured using 2-photon microscopy in Emx1-Ai93 mice. After mapping cortex, we placed the mouse under a 2-photon microscope, directing the field of view to the V1-LM border. Before performing 2-photon measurements, we generated a local widefield retinotopic map through the microscope objective using an LED and camera (*Figure 7A–D*), which confirmed that we had located the border region. By mapping through the microscope objective, we ensured that widefield and 2-photon measurements were aligned.

We then measured the receptive fields of layer 2/3 pyramidal neurons in the same field of view using 2-photon excitation and a sparse noise visual stimulus. In the example illustrated in *Figure 7*, we identified 366 neuronal somata in layer 2/3 (*Figure 7F*) and summing three experiments, identified 1276 somata. Neuropil contamination, the presence of fluorescence from surrounding GCaMP-labeled processes in the somatic pixels, is a characteristic of densely labeled tissue. We extracted

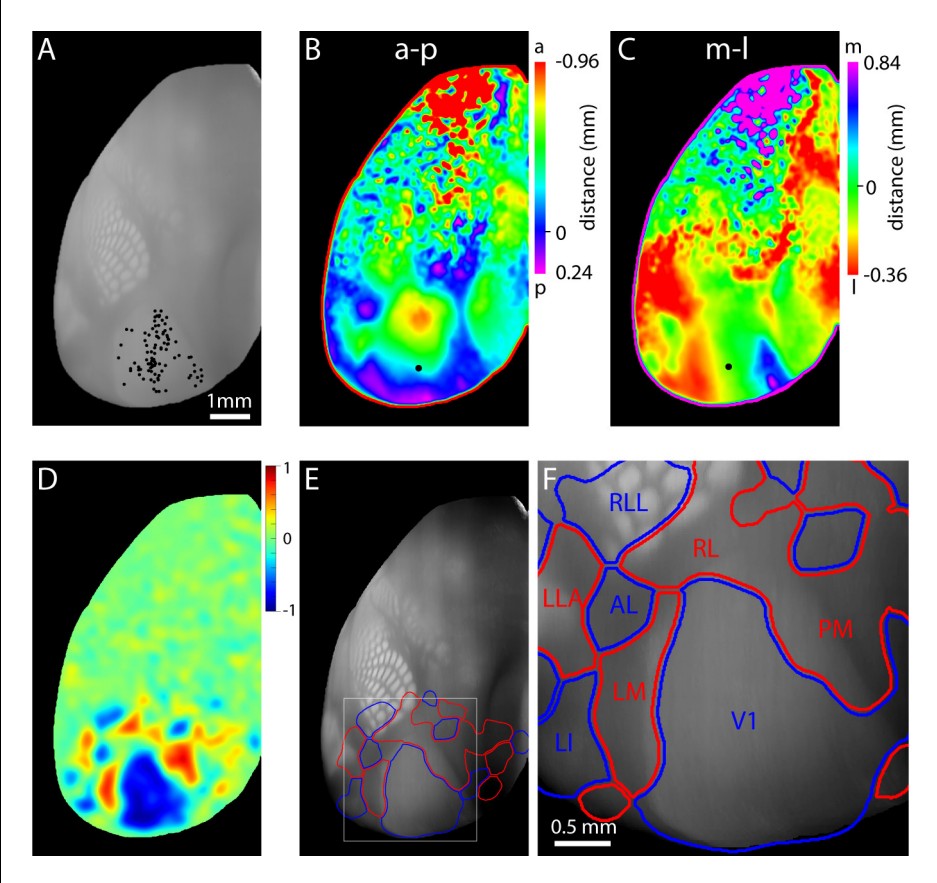

**Figure 6.** Retinotopy of projections from V1. (**A**) Locations of injections into V1 in 99 mice, selected from the Allen Brain Connectivity Atlas. Each point indicates an injection. Injection locations were registered to a 3D model of the mouse brain generated from 1675 brains and injections are illustrated on a top projection of the mean autofluorescence from the 3D model. Variations in autofluorescence clearly delineate major architectonic boundaries, including barrels in primary somatosensory cortex, primary auditory cortex, primary visual cortex and retrosplenial cortex. (**B** and **C**) Projection-based maps of connectivity with ipsilateral V1. Colors indicate the distance from the geometric center of V1 (black circle) from which the strongest projection arises, along the anterior (a) – posterior (p) and medial (m) – lateral (l) axes of V1. These maps are projection-based homologues of the azimuth and altitude maps generated from functional mapping (*Figure 4E,F*). (**D**) Projection sign map generated from the maps in panels **B** and **C**. (**E**) Automated borders generated from the projection sign maps of panel **D**, overlaid onto the autofluorescence top projection from panel **A**. (**F**) Subregion corresponding to the box in panel **E**. A lower threshold was used to generate the visual area borders from the sign map, eliminating the gap between areas V1, AL, RL and LM.

The following figure supplement is available for figure 6:

**Figure supplement 1.** Generation of projection-based retinotopy maps.

fluorescence from the neuropil surrounding each neuron and, as expected, found that these local regions of neuropil displayed receptive fields (*Figure 7—figure supplement 2*). Many of the neuropil regions displayed similar retinotopy to their parent somata. To correct for neuropil contamination, for each soma we measured and subtracted neuropil fluorescence from the surrounding pixels. Numerical simulations indicated that this approach was effective in removing neuropil contamination from somatic fluorescence measurements and confirmed that the map of somatic retinotopy was not an artifact of neuropil contamination (*Figure 7—figure supplement 2*).

Most somata displayed On and Off receptive subfields after neuropil subtraction (e.g. *Figure 7—figure supplement 1*). Further analysis was performed on cells with a receptive field with a maximum

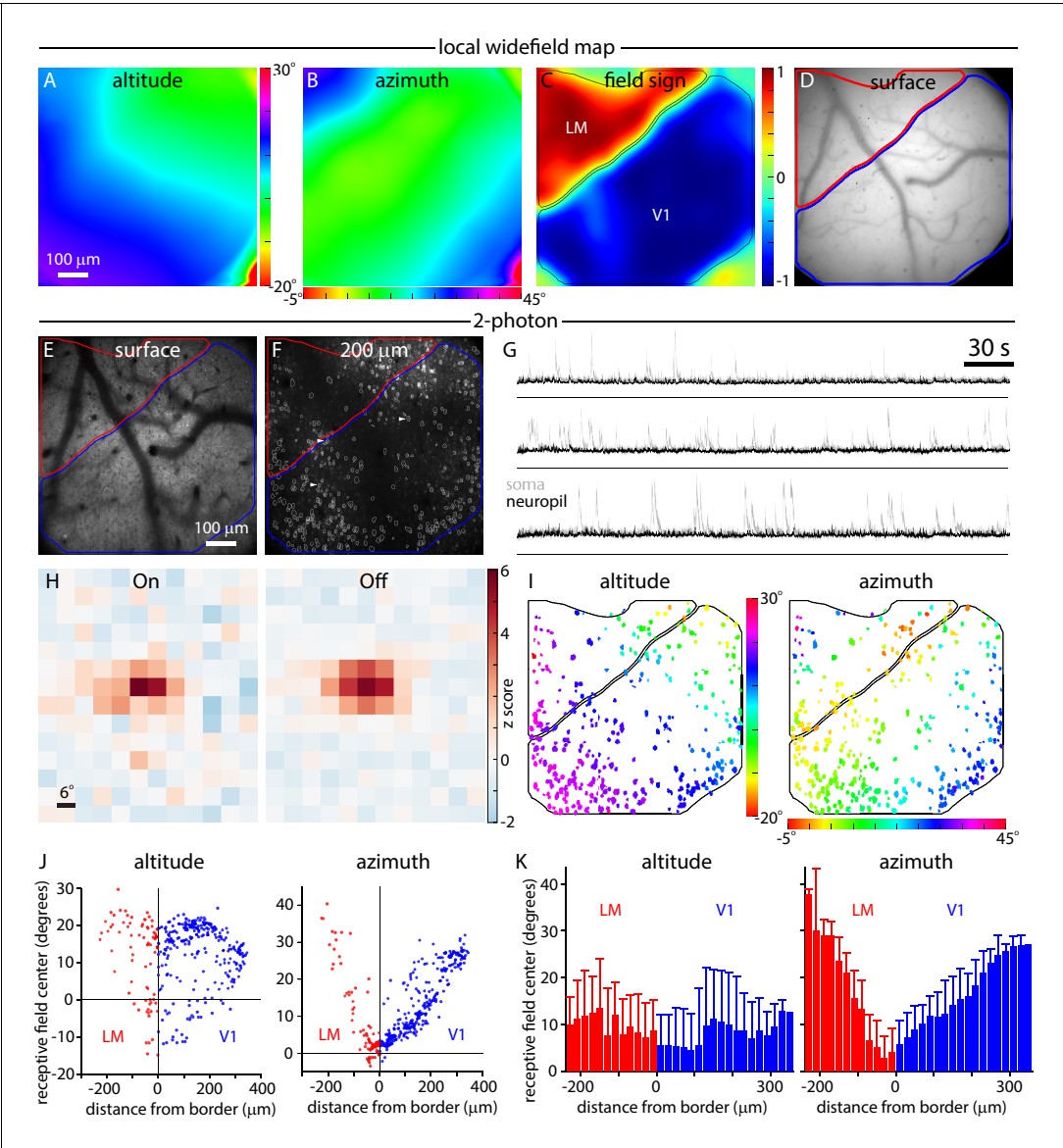

**Figure 7.** Widefield borders match single-cell retinotopy at the V1-LM border. (**A, B** and **C**) Local altitude and azimuth and field sign maps generated under the 2-photon microscope by widefield imaging through the x16 objective. Images were acquired with the objective focused 200 µm below the pial surface of cortex. (**D** and **E**) Images of surface vasculature acquired with the objective focused on the pial surface. Borders of V1 and LM derived from the field sign map are marked in blue and red, respectively. (**F**) 2-photon fluorescence image acquired with the microscope objective focused 200 µm below the pial surface of cortex. Borders of V1 and LM derived from the field sign map are marked in blue and red, respectively. White outlines indicate 366 somatic ROIs. (**G**) Example fluorescence traces extracted from the three somatic regions (before neuropil subtraction) marked with arrowheads in panel **F** (grey traces) and the corresponding neuropil regions (black traces). Fluorescence scale is in arbitrary units. A black horizontal line indicates zero fluorescence for each pair of traces. (**H**) On and Off receptive fields from an example cell, extending from −6 to 60 degrees in azimuth and −24 to 54 degrees in altitude. (**I**) Altitude and azimuth maps (from summed receptive fields) for the experiment illustrated in panel **F**. The color of each soma represents its receptive field center location. Of the 366 somata identified in this field of view, 336 displayed significant receptive fields (maximum z-score ≥ 2) and are illustrated in panel **I**. Black lines mark the borders of V1 and LM. (**J**) Plots illustrating the distribution of single-cell altitude and azimuth as a function of minimum distance to the V1-LM border. Each point represents a single soma from the field of view illustrated in panel **F**. (**K**) Single-cell altitude and azimuth as a function of distance to the V1-LM border. Results from three experiments were pooled, yielding 964 somata with receptive fields. Each bar represents the mean and standard deviation of cells binned by distance from the border, in 20 µm bins.
The following figure supplements are available for figure 7:

**Figure supplement 1.** On and Off receptive fields for an example cell.

*Figure 7 continued on next page*

*Figure 7 continued*

**Figure supplement 2.** Neuropil tuning and subtraction.

z-score $\geq$ 2, which included 92% of neurons (336 of 366) in the example in *Figure 7* and 76% (964 of 1276) of neurons across three experiments. For each neuron, we summed On and Off subfields and determined the center of the summed receptive field, then created somatic altitude and azimuth maps (*Figure 7I*). Receptive field centers formed an orderly map of retinotopy with a progression of altitudes along the V1-LM border and azimuth changing parallel to the V1-LM border. Importantly, the gradient in somatic azimuth reversed at the border measured by widefield mapping.

For most neurons, somatic altitude and azimuth were similar to the altitude and azimuth in the local widefield maps (*Figure 7A,B*). For a more quantitative comparison, for each neuron we plot somatic altitude and azimuth as a function of distance from the V1-LM border (defined by widefield fluorescence; *Figure 7J*). Somata with zero azimuth were close to the border (distance = 0) and azimuth increased approximately linearly with distance from the border, into V1 and into LM. As expected, the gradient of the relationship between somatic azimuth and distance to the border was steeper in LM than in V1, indicating greater cortical magnification in V1 than in LM near the border.

Finally, we pooled results from three experiments, calculating the mean somatic altitude and azimuth as a function of distance from the V1-LM border in 20 μm bins. The relationship displayed an orderly progression of somatic azimuth with distance from the border (*Figure 7K*). The lowest-azimuth bin was centered at −30 μm, indicating that the single-cell border was 30 μm lateral of the widefield retinotopic border. This 30 μm difference in retinotopic border locations is insufficient to account for the 100–300 μm mismatch between retinotopic and architectonic border locations.

## Arrangement and visual coverage of higher visual areas in the mouse

In addition to the new field sign patches and medial displacement of the lateral border of V1, GCaMP6 fluorescence-based retinotopic maps differ from previous maps in two respects. Firstly, area P extends across the posterior border of V1, from LM to PM. Secondly, area M is displaced medially relative to the location reported in *Garrett et al. (2014)*. One result of these differences is an almost continuous ring of field sign positive areas surrounding V1, broken only by the field sign negative area AM.

To assess the representation of visual space in V1 and of the surrounding extrastriate regions in the mouse, we plot the coverage of these regions in retinotopic coordinates (*Figure 8*). V1 included representation of the right visual hemifield between ~0 (the vertical meridian) and 90° azimuth and ~25–35° above and below the horizontal meridian (*Figure 8A*). The 4 positive field sign patches around V1 (LM, RL, PM, P; *Figure 8A*) each represented a portion of the visual field and each was biased towards a different quadrant: LM was biased towards the upper nasal visual field, RL the lower nasal field, P upper temporal field and PM lower temporal field (*Figure 8C*). The summed coverage of these four patches approximated that of V1 with modest overlap in coverage in two narrow strips of visual space ~0–15° above the horizon and ~40–50° from the vertical meridian. The intersection of these zones of overlap included the center of coverage of V1 (altitude 7.4 ± 2.1 degrees, azimuth 37.8 ± 1.4 degrees, 14 mice), which was represented in areas LM, RL and PM. This overlap was also observed in maps of eccentricity, where a representation of the center of visual coverage was present for each area (*Figure 9B*).

Like the ring of four positive field sign patches, negative field sign patches LI, AL and AM exhibited coverage that overlapped at the center of coverage of V1 (*Figure 8B*). LI represented the upper visual field at ~30–40° azimuth; AL represented the nasal visual field at ~0–15° altitude; and AM represented the lower visual field at 10–30° azimuth (*Figure 8B*).

Of the field sign patches newly identified by GCaMP6 fluorescence mapping, the positive field sign patches MMA and in retrosplenial cortex exhibited the largest coverage, extending in a ~10° strip ~30–40° along and ~5–10° above the horizontal meridian, respectively, and the negative field sign patches RLL and MMP each represented ≤10° x 20° of the visual field (*Figure 8C*). Like other patches towards the periphery of the mouse visual cortex (areas POR and LLA; *Garrett et al.,*

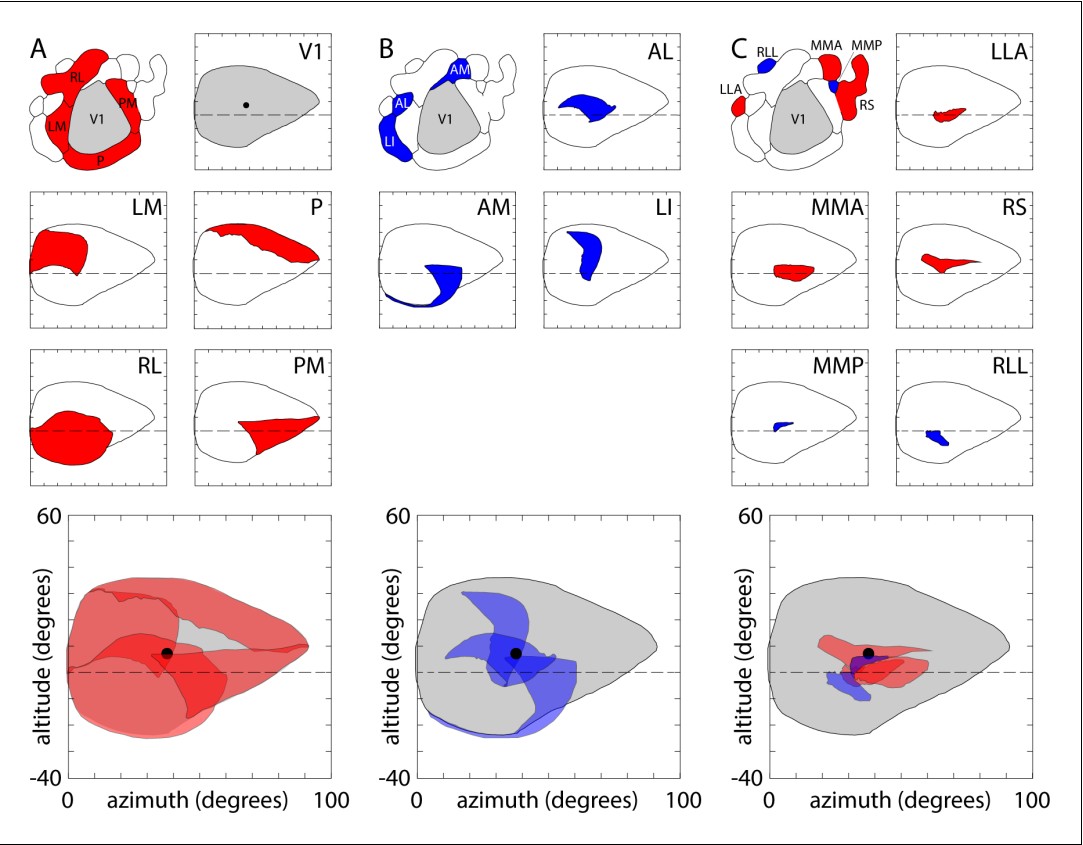

**Figure 8.** Visual coverage across areas. (**A**) Coverage map of visual space by V1 and surrounding positive field sign patches. Top left, overview map of V1 (grey) and the four surrounding positive field sign patches (red; LM, RL, PM, P). Top right, V1 coverage map. Locations represented in V1 are indicated in grey. Circle indicates the center of coverage of V1 at 7.4° altitude, 37.8° azimuth. Dashed line indicates the horizontal meridian. Center panels, coverage maps of positive field sign patches that border V1 (LM, RL, PM, P). Each coverage map illustrates the one positive field sign patch (in red), overlaid on the coverage of V1 (black outline). Note that coverage of PM excludes retrosplenial cortex. Lowest panel, overlapping coverage of 5 areas (V1 in grey; LM, RL, PM, P in red). (**B**) Coverage of negative field sign patches LI, AL and AM. Coverage of each patch (blue) is overlaid on the coverage of V1 (black outline). (**C**) Coverage of remaining positive and negative field sign patches.

The following figure supplement is available for figure 8:

**Figure supplement 1.** Expanded coverage.

*2014*), the newly-identified field sign patches each represented smaller regions of the visual field around the center of coverage of V1.

The limited coverage of most field sign patches is also visible in eccentricity maps, in which representation of the peripheral visual field is limited to V1 and its immediate neighbors (*Figures 8A* and *9B*). Furthermore, neighboring field sign patches generally share a similar bias, with the lower visual field represented primarily in anterior field sign patches and upper visual field in posterior patches, the nasal visual field primarily in lateral patches and temporal visual field in medial patches (*Figure 9C–F*). In short, almost all higher visual areas in the mouse display a strong bias in representation which is shared with their immediate neighbors, but all share a representation of the center of coverage which corresponds approximately to the center of gaze.

## Discussion

Our results establish GCaMP6 fluorescence as a tool for generating retinotopic maps of mouse cortex. The exquisite signal-to-noise ratio of GCaMP6 permitted rapid mapping and also revealed several new field sign patches, enabling us to map the medial extent of visual cortex. Adding the four

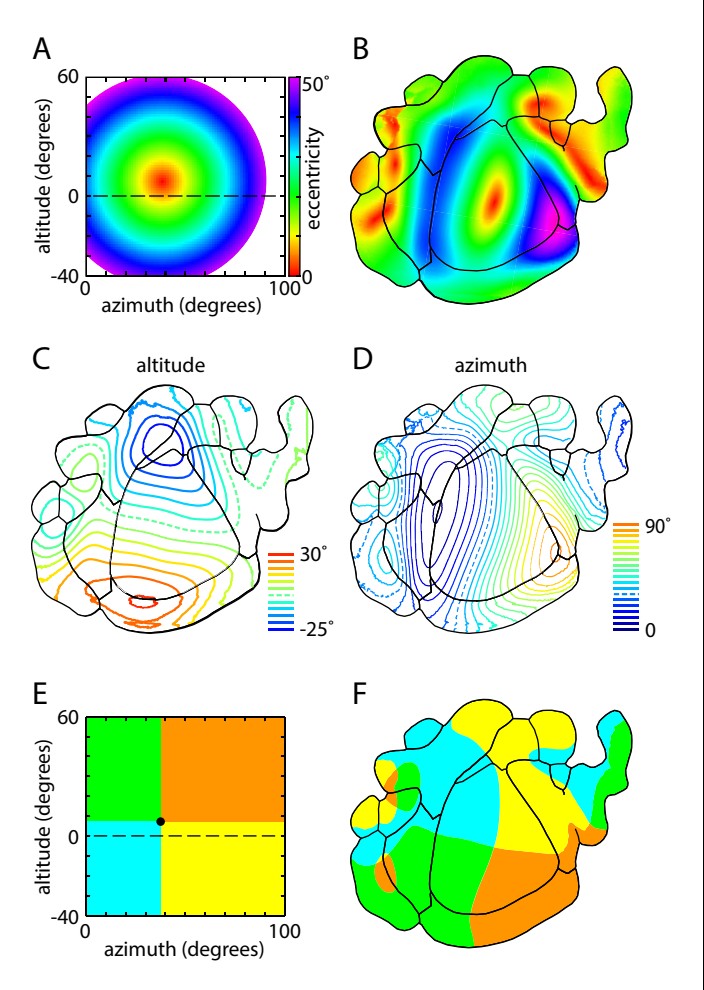

**Figure 9.** Magnification and representation of visual space across visual cortex. (**A**) Color map of visual space in eccentricity coordinates. Color (bar) indicates the distance (in degrees) from the center of coverage of V1 (7.4 ± 2.1 degrees altitude, 37.8 ± 1.4 degrees azimuth). (**B**) Eccentricity map of mouse visual areas, using the color scheme indicated in panel **A**. (**C,D**) Altitude and azimuth contour plots of mouse visual cortex, overlaid on mean field sign borders. Dashed line represents horizontal meridian. Contours are at 5° intervals from −25 to 30° in altitude and 0 to 90° in azimuth. (**E**) Colored sector map of visual space, with the division between upper and lower visual fields at 7.4° altitude, 37.8° azimuth, which corresponds to the center of coverage of V1. (**F**) Colored sector map of mouse visual cortex, with colors corresponding to those in panel **B** and denoting representation of the four quadrants of the visual field.

new field sign patches (MMA, MMP, RLL, RS) to 12 published areas (V1, LM, LI, AL, LLA, RL, A, AM, PM, M, P, POR), mouse cortex contains 16 distinct retinotopically-organized regions.

## Extension of retinotopic organization

Our results indicate that retinotopic organization is more extensive in mouse cortex than previously appreciated. We observed five instances of extended retinotopy. Firstly, area P extends across the posterior extent of V1. Tracer injections into V1 have revealed projections into this region in rat and mouse (*Olavarria and Montero, 1981*, *1984*; *1989*; *Coogan and Burkhalter, 1993*; *Wang et al., 2012*) and recent maps have suggested the presence of one or more positive field sign patches posterior to V1 (*Garrett et al., 2014*). The signal-to-noise ratio of our measurements may be particularly limited in this posterior region, along the edge of our cranial window. Furthermore, at its posterior extent cortex is folded and abuts the transverse sinus, and there may be additional retinotopic

structure under the sinus or in the folded region that we were unable to access. This additional region may include representation of the center of coverage of V1, which is absent from our map of coverage of P.

We identified two field sign patches in the anterior and posterior aspects of MM, the narrow strip of visual cortex immediately lateral to retrosplenial cortex (*Wang and Burkhalter, 2007*). These two patches were labeled MMA and MMP. Variations in the density of neurons immunoreactive for non-phosphorylated neurofilament protein across medial visual areas have led to the suggestion that MM contains multiple distinct regions (*Van der Gucht et al., 2007*). Furthermore, MM receives direct projections from V1 that terminate in distinct anterior and posterior regions (*Wang and Burkhalter, 2007*) and likely correspond to MMA and MMP.

Retinotopy extends into retrosplenial and barrel cortices. Retinotopic organization presumably facilitates the processing of multimodal information in these areas (*Olcese et al., 2013*). For example, like RL (*Garrett et al., 2014*), RLL is biased towards the nasal, lower field and may convey to barrel cortex visual information on whisker location and nearby objects. Similarly, the presence of a visual map in retrosplenial cortex is consistent with its visual responsiveness (*Murakami et al., 2015*) and its role in spatial memory and navigation (*Vann et al., 2009*).

## Limitations of mapping with population imaging and simple visual stimuli

We used a flickering checkerboard to drive cortical activity during retinotopic mapping, but some areas might be activated more effectively, and mapped more readily, with other visual stimuli. In primates, for example, neurons in higher visual areas are more strongly activated by complex stimuli (*Nassi and Callaway, 2009*). In the mouse, some higher areas are preferentially activated by high or low spatial and temporal frequencies (*Marshel et al., 2011*; *Andermann et al., 2011*; *Roth et al., 2012*; *Tohmi et al., 2014*), leading to the suggestion that simple stimuli with different spatial and temporal frequency characteristics might preferentially activate different regions of cortex. However, the checkerboard pattern includes sharp borders between black and white regions of the checkerboard that naturally include a broad band of spatial and temporal frequencies. Furthermore, retinotopically-organized regions occupy most of the territory between primary sensory areas in posterior cortex, suggesting that there are few additional visual areas to be discovered in the mouse. As a result, we would not expect substantial differences in maps of higher visual areas generated with checkerboards with distinct spatial or temporal frequency band characteristics.

Our maps are derived from population imaging, which naturally imposes limits on the resolution with which we can map areas and borders. Furthermore, imaging of widefield fluorescence signals at the brain surface is strongly contaminated by vasculature artifacts, which we address by focusing below the surface of cortex. Defocusing has a blurring effect and can limit effective resolution. Widefield fluorescence signals in Emx1 mice are presumably the average of fluorescence from many neurons, with dendrites and axons providing fluorescence signals far from the soma. Processes probably cross regional borders, making it likely that dendritic and axonal signals from neurons on one side of the border contribute to fluorescence on the other side of the border. It is unclear the extent to which border-crossing processes blur borders since blurring will depend on many factors, including whether stimulus-locked changes in the fluorescence of processes more closely match stimulus-locked changes of the parent soma or of the local network into which the processes extend. Given the potential for blurring of borders, it is perhaps surprising that widefield and 2-photon images give border locations which are separated by only a few tens of micrometers.

## Cortical regions, field sign patches and visual areas

The cellular and laminar origins of visually-evoked changes in widefield GCaMP6 fluorescence remain unknown. GCaMP6 is likely expressed throughout excitatory neurons, including their axons, and it is possible that some of the observed retinotopic organization, such as that in retrosplenial and barrel cortices, results from retinotopically-organized axons. Indeed, the lack of retinotopy in retrosplenial and barrel cortices of Rorb-Ai93 mice suggests a lack of somatic retinotopy in layer four and upper layer five in these regions.

We have used the term 'patch' to refer to regions of cortex identified by our analysis routine. We define patches as regions of cortex with retinotopic organization that are distinct from neighboring

patches, either because of a reversal of chirality in the visual map or because the neighboring patches contain redundant representations of visual space. We use the term 'patch' to avoid implying any specific mechanistic basis for the retinotopic organization, such as somatic retinotopy. We have used 'region' simply to refer to part of cortex, with no intended implications regarding structure or properties.

'Visual area' is a more established term that implies the consistent identification of borders with multiple approaches, including anatomical and functional measurements (*Orban et al., 2004*; *Wandell et al., 2007*). Furthermore, we consider somatic retinotopy necessary in a visual area. In the absence of evidence of somatic retinotopy and supporting evidence from other techniques, we have, where possible, avoided using 'area' to refer to some patches.

## Retinotopic and architectonic borders

In maps of mouse visual areas drawn with the assistance of architectonic boundaries, the borders of V1, AL, LM and RL all intersect at a common point (e.g. *Wang and Burkhalter, 2007*). Our results indicate that in retinotopic maps obtained with functional imaging and with projection-based mapping, the lateral border of V1 is more medial than the architectonic boundary. As a result, V1 and AL lack a common border. The territory between V1 and AL is occupied by RL and LM, which share a border extending ~100–300 μm. The architectonic border of V1 runs through RL and LM and typically intersects the medial tip of AL. Hence the medial tip of AL intersects the lateral border of architectonically-defined V1, but not retinotopically-defined V1. The border mismatch helps explain why the relative positions of visual areas, and their points of intersection, can differ on maps based on results from different techniques.

The mismatch between the architectonic and retinotopic borders of V1 is consistent in our results across three methods, indicating that the architectonic border of V1 may correlate with functions other than retinotopic reversal. One possibility is that the cytochrome oxidase-rich region includes the binocular region around the lateral border of V1, which would help explain the mismatch since areas RL and LM include representations of the binocular zone (*Wagor et al., 1980*). Another possibility is that there is a thin, largely monocular zone along the medial edge of LM and RL which is cytochrome oxidase-rich (*Laing et al., 2015*). It is unclear whether there is a mismatch in the architectonic and retinotopic borders of other visual areas, such as between LM and LI (*Wang et al., 2011*) and further studies will be required to establish the similarity of architectonic and retinotopic border locations for other visual areas.

## Representation of visual space and the organization of mouse visual areas

A prominent feature of GCaMP6-based retinotopic maps is the four positive field sign areas (RL, LM, P and PM) neighboring V1 that form an almost continuous ring, broken only by the negative field sign area AM. This arrangement resembles early visual areas in primates, in which V1 is a negative field sign region surrounded by V2, and V2 is a continuous strip of positive field sign tissue broken only at a single point (anterior tip of the calcarine sulcus; *Gattass et al., 2005*; *Sereno et al., 1995*; *Warnking et al., 2002*; *Silver and Kastner, 2009*; *Wilms et al., 2010*; *Laumann et al., 2015*).

Early studies of mouse visual cortex revealed a retinotopic map in V1 and a second map anterior and lateral to V1, leading this second region to be named V2 (*Wagor et al., 1980*). Further studies revealed several overlapping maps within mouse and rat V2, which was therefore split into several named visual areas, including LM and RL (*Olavarria and Montero, 1981*, *1984*, *1989*; *Malach, 1989*; *Olavarria et al., 1982*; *Thomas and Espinoza, 1987*; *Coogan and Burkhalter, 1993*; *Wang and Burkhalter, 2007*). Subsequent studies have documented additional distinctions between the four positive field sign areas neighboring V1. For example, a study of the laminar termination patterns of axonal projections between cortical areas suggested that the areas around V1 occupy different positions in the hierarchy of visual areas in rats (*Coogan and Burkhalter, 1993*). Consistent with this suggestion, medial and lateral regions of visual cortex, likely corresponding to LM, RL and PM, display different response latencies in mice (*Polack and Contreras, 2012*), and neurons in these areas differ in their mean receptive field sizes (*Wang and Burkhalter, 2007*) and in their tuning to the spatial and temporal characteristics of visual stimuli (*Andermann et al., 2011*; *Marshel et al., 2011*; *Roth et al., 2012*; *Glickfeld et al., 2013*; *Tohmi et al., 2014*). These studies all support the

identification of multiple visual areas in the tissue bordering V1, in contrast with primates where the tissue neighboring V1 is considered a single visual area (V2).

Our results indicate that the visual field representations of the four areas around V1 (LM, P, PM, RL) are complementary, with the outline of the coverage of these four areas matching the outline of V1 coverage. Our results likely underestimate the range of visual coverage in each area and the overlap between areas. V1 included a visual coverage range of ~60° in altitude and ~90° in azimuth, comparable to measurements with reflectance-based imaging (e.g. *Garrett et al., 2014*) but smaller than the range measured with single-cell electrical recordings (~100° in altitude and ~150° in azimuth; *Wagor et al. (1980)*). Single-cell recordings are likely to provide a larger range for two main reasons.

Firstly, widefield mapping identifies retinotopic position based on the peak of activity, thereby emphasizing the receptive field center. The outer limits at which visual stimuli can evoke activity in V1 will be expanded, by approximately the radius of the largest receptive fields of neurons at the edges of V1. A more pronounced expansion in effective coverage is expected in higher visual areas, where receptive field sizes are larger (*Wang and Burkhalter, 2007*). Expanded coverage, calculated using published receptive field sizes (*Wang and Burkhalter, 2007*) is illustrated in *Figure 8—figure supplement 1*. For V1, expanded coverage includes ~70° in altitude and ~100° in azimuth. Mean local receptive field size may change across a visual area, which possibly accounts for the mismatch of coverage at the borders of neighboring areas (e.g. the V1-LM border, where the outer limits of the estimated V1 and LM coverage differ).

Secondly, widefield fluorescence measures the average retinotopy of the local population of neurons. Single-cell receptive field centers vary around the mean retinotopy. For example, in V1 most pyramidal neuron receptive fields are within ~7 degrees of the local population average (*Bonin et al., 2011*). How variability relative to the mean local receptive field center changes towards the borders of V1 or in other visual areas is unknown, but an expansion of coverage of 7 degrees at each border (in addition to the expansion due to receptive field size) would result in total coverage of V1 of ~85° in altitude and ~115° in azimuth, closer to the numbers from single-cell recordings. These differences between population and single-cell coverages may explain the small apparent regions of coverage of many higher visual areas, in which receptive field sizes and perhaps also the scatter in receptive field centers are greater than in V1. Even allowing for some expansion of coverage for each visual area, our results indicate that no higher visual areas (even those immediately surrounding V1) contain a complete description of the visual hemifield. Hence information on features that subtend more than approximately a quarter of the visual field will be routed to different early visual areas, with the result that information on features in different locations within a single visual scene will be processed in regions with different functional properties. Further studies will be needed to understand the advantages and limitations of processing information from different locations in visual space in functionally distinct regions of visual cortex.

## Materials and methods

### Transgenic mice

In this study we employed six mouse lines:

- Ai95(RCL-GCaMP6f): B6;129S-*Gt(ROSA)26Sor*$^{tm95.1(CAG-GCaMP6f)Hze}$/J (Jax stock number 024105; *Madisen et al., 2015*)
- Ai96(RCL-GCaMP6s): B6;129S6-*Gt(ROSA)26Sor*$^{tm96(CAG-GCaMP6s)Hze}$/J (Jax stock number 024106; *Madisen et al., 2015*)
- Ai93(TITL-GCaMP6f): *Gt(ROSA)26Sor*$^{tm5(ACTB-tTA)Luo}$ *Igs7*$^{tm93(tetO-GCaMP6f)Hze}$/HzeJ (Jax stock number 024107; *Madisen et al., 2015*)
- Emx1-IRES-Cre: B6.129S2-*Emx1*$^{tm1(cre)Krj}$/J (Jax stock number 005628; *Gorski et al., 2002*)
- Rorb-IRES2-Cre (Jax stock number 023526; Allen Mouse Brain Connectivity Atlas: *Oh et al., 2014*; *Harris et al., 2014*)
- CaMK2a-tTA: B6.Cg-Tg(Camk2a-tTA)1Mmay/DboJ (Jax stock number 007004; *Mayford et al., 1996*).

Ai93, Ai95, and Ai96 are floxed GCaMP6 reporter lines (*Madisen et al., 2015*), which differ in the promoter/enhancer used to drive GCaMP6 expression and in the isoforms of GCaMP6 expressed.

Ai95 and Ai96 lines employ a ROSA-CAG promoter to drive Cre-dependent expression of GCaMP6f and GCaMP6s, respectively (*Madisen et al., 2015*). Stronger expression of GCaMP6f is achieved in Ai93, using the TIGRE promoter.

For most experiments reported here, these reporter lines were crossed with Emx1-IRES-Cre mice to drive expression in pyramidal neurons throughout neocortex. Ai95 and Ai96 express GCaMP6 in the presence of Cre and all experiments were performed on double transgenic mice hemizygous for Cre and for GCaMP6. As Ai93 requires the presence of both Cre and tTA to drive expression, Ai93 mice were crossed with Emx1-IRES-Cre and CaMK2a-tTA to yield triple transgenic mice that were hemizygous for all three genes. We refer to these crosses as Emx1-Ai93, Emx1-Ai95 and Emx1-Ai96 mice. For a small sub-set of experiments (*Figure 5J–L*), Ai93 was crossed with Rorb-IRES2-Cre and CaMK2a-tTA to yield Rorb-Ai93 mice. Both sexes of mice were used and all mice were maintained on a B6/C57 background.

## GCaMP6 expression

The pattern of GCaMP6 expression was examined in two mice of each genotype at postnatal day 67–114 (Emx1-Ai95 P75, Emx1-Ai96 P67, Emx1-Ai93 P67, wild-type P114). Brains were fixed by transcardial perfusion with 4% (w/v) paraformaldehyde. 100 µm-thick coronal sections were cut using a vibratome and mounted in Vectastain. Endogenous GCaMP6 was imaged by widefield or 2-photon fluorescence microscopy. Image analysis was performed in ImageJ.

## Surgery

Retinotopic imaging was performed through a 5 mm diameter circular cranial window positioned over visual areas of the left hemisphere. The preparation was similar to that described previously (*Andermann et al., 2010*; *2011*). Briefly, under isoflurane anesthesia, a head restraint bar was attached to the skull using C & B Metabond (Parkell) and a 5 mm craniotomy opened at center coordinates 2.7 mm lateral, 1.3 mm anterior to lambda. The craniotomy was sealed with a stack of three #1 coverslips, attached to each other using optical adhesive (Norland) and to the skull with Metabond. The mouse was permitted to recover for at least seven days and conditioned to the head restraint and running wheel for several days before mapping.

All experiments and procedures were approved by the Allen Institute Animal Care and Use Committee.

## Widefield microscopy

Widefield fluorescence images were acquired with a 1:1 optical relay using two x1 PlanAPO dissecting microscope lenses (Leica, 10450028). Illumination was from a blue LED (M470, Thorlabs), via a bandpass filter (469/35, Semrock) and fluorescence was detected by a CCD camera (Orca R2, Hamamatsu) via a 497 nm dichroic and 525/39 bandpass filter (Semrock). Parts were mounted on a macroscope (THT scope, Scimedia) with its optical axis tilted 22 degrees in the coronal plane such that the optical axis was perpendicular to the cranial window. The focal plane of the microscope was positioned deep in cortex, thereby defocusing the surface vasculature during retinotopic mapping. Illumination and image acquisition were controlled with software written by JW using the Hamamatsu Video Capture Library for Labview, v.2.0.2.

## Mapping procedure and visual stimuli

During imaging, the head was restrained via the implanted bar and the eyes were on a horizontal plane. Visual stimuli were displayed on a 40" LED TV (Samsung 6300), placed 13.5 cm from the right eye. The mouse was oriented with its midline at ~30° to the plane of the monitor. Visual coordinates were calculated with respect to the midline (azimuth coordinates) and the horizontal plane through the eyes (altitude coordinates). The monitor covered approximately −10 to 130 degrees in azimuth and −50 to 60 degrees in altitude. The luminance of the stimulus monitor ranged from 0.05 (black) to 177 (white) cd/m$^2$.

Awake mice were free to run on a 16.5 cm diameter disk. For imaging under anesthesia, mice were anesthetized with <1% isoflurane (inhaled) and chlorprothixene (2.5 mg/kg, intramuscular) and silicon oil (10,000 molecular weight) was applied to both eyes to prevent dehydration.

Retinotopic maps were generated by sweeping a bar across the monitor (*Kalatsky and Stryker, 2003*). The bar contained a flickering black-and-white checkerboard pattern, with spherical correction of the stimulus to stimulate in spherical visual coordinates using a planar monitor (*Marshel et al., 2011*; *Garrett et al., 2014*; *Videos 2* and *3*). The pattern subtended 20 degrees in the direction of propagation and filled the monitor in the perpendicular dimension. The checkerboard square size was 25 degrees. Each square alternated between black and white at 6 Hz. To generate a map, the bar was swept across the screen ten times in each of the four cardinal directions, moving at nine degrees per second. To ensure that stimulus-evoked activity had subsided between sweeps, a gap of $\geq$5 s was inserted between sweeps, resulting in repetition of the stimulus at 0.048 Hz for vertically-moving stimuli and 0.043 Hz for horizontally-moving stimuli.

During mapping, fluorescence images were acquired at 10 Hz with 2×2 binning, resulting in an effective pixel size of 12.9 µm at the sample. Across different mouse lines, fluorescence varied ~100 fold for each mouse and illumination intensity was adjusted to almost fill the camera well depth during periods of activity. Mean ± SEM illumination intensity for GCaMP mice was 89 ± 21 µW/mm$^2$ (range 19–210 µW/mm$^2$).

A slight decline in fluorescence was generally observed during mapping, presumably due to photobleaching. Photobleaching was approximately linear with time and intensity, with fluorescence declining at 235% J$^{-1}$mm$^2$ (11 mice). At the mean illumination intensity during mapping (89 µW/mm$^2$), resting fluorescence declined at a mean rate of 0.02% per second or 1.25% per minute.

Brief stimuli (*Figure 1*) consisted of a white circle 20° in diameter, 50 ms in duration, centered at 60° azimuth and 0° altitude, on a black background. The white circle was displayed 20 times at 0.2 Hz. Images were acquired under 1.8 µW/mm$^2$ illumination at ~64 Hz with 8×8 on-chip binning, resulting in an effective pixel size of 51.6 µm at the sample.

## Widefield image analysis

All image analyses were performed in the Python programming environment, with OpenCV/SimpleCV libraries, after subtraction of a camera bias of 100 digitizer units.

For retinotopic mapping experiments, our analysis followed the methods described by *Garrett et al. (2014)* with minor modifications. We first created △F movies: for each presentation of the checkerboard stimulus, from all frames of the movie we subtracted an image corresponding to the mean of the 2 s before the start of stimulation. We then created a stimulus-triggered mean △F movie for each of the four stimulus directions, averaging 10–40 trials in each direction. To generate azimuth and altitude position maps, from fluorescence versus time data for each pixel we extracted retinotopic positions from the phase of the first harmonic component of the Fourier series, with peak frequencies of 0.043 Hz (0.022–0.065 Hz band) for azimuth and 0.048 Hz for altitude (0.024–0.072 Hz band) maps, corresponding to the periodicity of the stimulus (moving at 9° per second). To cancel the delay from stimulus to response, we calculated a pixel-by-pixel average of the visual response positions derived from movies for stimuli traveling in opposite directions (for altitude, we averaged bottom-to-top and top-to-bottom movies; for azimuth, nasal-to-temporal and temporal-to-nasal movies).

Azimuth and altitude maps were combined to generate a visual field sign map (*Sereno et al., 1994*; *1995*; *Garrett et al., 2014*), where the visual field sign at each pixel is the sine of the angle between the local gradients (derived with the numpy.gradient function) in azimuth and altitude. The visual field sign map was converted into borders as described by *Garrett et al. (2014)*, *Juavinett et al. (2016)*, as outlined in *Figure 2—figure supplement 1*. The visual field sign map was spatially filtered with a Gaussian kernel (standard deviation range 6–10 µm, mean ± SEM of 8.14 ± 0.08 µm, 14 mice) then thresholded to create a binary mask. For each map the threshold was tuned manually over a narrow range (field sign values of 0.2–0.4, mean ± SEM of 0.32 ± 0.004, 14 mice). Each suprathreshold patch was dilated to yield a border width between patches of one pixel. Isolated pixels were eliminated with open/close operations. The binary mask was converted into an initial 'raw' patch map in which each pixel value was −1, 0 or 1. Patches were further processed with a split/merge routine in which patches with >10% redundancy in visual coverage were split using a watershed routine at the local minimum of the visual eccentricity map and, subsequently, adjacent patches with the same sign and <10% redundancy in visual coverage were merged. Patches smaller than 0.00166 mm$^2$ (100 pixels) were discarded.

Python code which accepts altitude and azimuth maps and identifies patch borders is available at https://github.com/zhuangjun1981/retinotopic_mapping and *Supplementary file 1*. The code includes an example data set and associated Python notebook illustrating each analysis step. Within the analysis routine, there are 13 variables that are set manually. Further explanation of these variables and the values employed in our analyses are stated in the code.

Results from multiple mice were pooled to create mean azimuth, altitude and field sign maps. Maps for each mouse were centered on the centroid of V1 and rotated to align the major axis of the azimuth gradient. Differences in relative positions of mouse and monitor were corrected by defining the V1, LM, RL border as 0° in altitude and azimuth, necessitating a correction of 0.3 ± 1.8 degrees in altitude and 14.9 ± 2.0 degrees in azimuth (14 mice). Mean azimuth and altitude maps were then calculated by vector summation (*Garrett et al., 2014*). To create patch borders for pooled results from the average sign map (*Figure 3B and C*), we thresholded the sign map at 0.3 and further processed the patches as described above (split, merge, discard small patches).

For brief, circular stimuli (*Figure 1F–J*) twenty trials were averaged and $F_{rest}$ was calculated as the mean of the 2 s pre-stimulus period. $\triangle F/F$ movies were generated by subtraction of $F_{rest}$ from all pixel intensities, before division by $F_{rest}$. Fluorescence was extracted from the region of the image with the greatest peak $\triangle F/F$ (which was generally V1): after spatial filtering with a Gaussian kernel ($\sigma$ = 258 μm, corresponding to five pixels) to reduce noise, a 516 μm (10 pixel) square region of interest was centered on the brightest pixel in the maximum intensity projection of the movie (dimmest pixel in wild-type mice, in which fluorescence declined with stimulation). Results in *Figure 1* report the fluorescence intensity within this region.

## Eye tracking

An image of the right eye was recorded by infrared (IR) imaging. The peri-ocular region was illuminated with an 850 nm LED (Ostar SFH4750) and an image of the right eye was reflected by a low-pass dichroic mirror (Semrock, FF750-SDi02-25×36) placed ~3 cm from the eye and between the eye and the stimulus monitor. Distance from the eye to the camera was ~30 cm. Pupil images were acquired via a Microsoft webcam chip through a Tamron CCTV lens (23FM50SP, focal length: 50 mm) and long-pass filter (Thorlabs FEL0800). Acquisition was at 30 Hz with 320×240 pixels, effective pixel size 18 μm.

To calculate pupil position and area, we first located and tracked the corneal reflection of the infrared LED (the brightest object in the image). The LED reflection was masked to prevent it from interfering with pupil detection (e.g. in the situation that the LED reflection was inside the image of the pupil). The image was then blurred and edges were detected and exaggerated using the OpenCV 'Canny edge detection' function (*Canny, 1986*). The pupil appeared as a dark, approximately round object. A region containing the pupil was manually selected and all spatially separate dark objects in the selected region were outlined. Each outline was subjected to an 'open' operation and all outlines with an area of less than 0.03 mm$^2$ were discarded. For each outline, centroid location, area, average intensity and roundness were calculated and rank order similarity to the pupil in the previous frame was calculated for each parameter. The outline with the smallest summed rank across all parameters was identified as the pupil.

Mice blinked periodically. The reflection of the LED was absent from frames in which the eyelid was closed, permitting automated identification of blink events. In the absence of a previous frame containing the pupil (the first frame of the movie and first frame after a blink) the pupil was identified as the largest, most circular dark outline, again determined with a minimum rank sum criterion.

After pupil identification, pupil area was calculated by counting the number of pixels within the borders of the extracted outline and multiplying by the area of a single pixel ($3.24 \times 10^{-4}$ mm$^2$).

Pupil motion in retinotopic coordinates (changes in gaze angle) was calculated under the assumption that the mouse eye is spherical, with a radius of 1.7 mm (*Remtulla and Hallett, 1985*). First, for each frame we determined the location of the centroid of the pupil and of the reflection of the IR-LED. From these values we calculated the position of the pupil (in pixels) relative to the LED reflection in horizontal and vertical planes ($X_i$ and $Y_i$, where i represents frame number). The cardinal axes of the camera chip, which was mounted parallel to the optical table, were used to define the horizontal and vertical planes of pupil movements. Pupil positions in pixels were converted to changes in azimuth and altitude gaze angles by trigonometry: $\Delta\theta_{azi}$ = arcsin($\Delta X_i/r$); $\Delta\theta_{alt}$ = arcsin($\Delta Y_i/$

r), where $\Delta X_i$ and $\Delta Y_i$ represent the deviation of X and Y position in the $i^{th}$ frame from the mean location during the movie, and r was set to be 1.7 mm (the average radius of mouse eye ball).

## Registration of retinotopic maps to chemoarchitectonic and cytoarchitectonic borders

Detailed protocol for registration can be found at Bio-protocol (*Zhuang et al., 2018*). Retinotopic maps were compared to chemoarchitectonic borders via cytochrome C oxidase (CO)-stained tangential sections of flattened cortex from 4 Emx1-Ai96 mice. For each mouse, the retinotopic map was aligned to the bright field image of the surface vasculature within the cranial window. The mouse was transcardially perfused, sequentially, with saline (10 ml/min for 10 min); 5 µg/ml DyLight 649-lectin conjugate (Vector Laboratory, 5 ml/min for 5 min, to label vascular endothelium); 5 min pause; 1% (w/v) paraformaldehyde (PFA) in PBS (5 ml/min for 20 min). After perfusion, a fluorescence image of labeled vasculature within the cranial window was acquired (640/690 nm excitation emission). The left cortex was isolated, flattened between glass slides, and post-fixed overnight in 4% PFA (*Wang and Burkhalter, 2007*). The fixed cortical sheet was cut into an asymmetrical shape to aid future alignment of fluorescence and brightfield images of the flattened tissue. A fluorescence image of the surface vasculature was acquired and the tissue was cut tangentially into 50 µm sections. Sections were stained for CO as described previously (*Tootell et al., 1988*; *Wang et al., 2012*).

Images acquired at different stages of tissue processing were sequentially aligned to the CO image by manual warping using the TrakEm2 plug-in in ImageJ. Key registration steps included the alignment of vasculature images across live, whole-mount and flattened preps (*Figure 4E*). Alignment of images of surface vasculature and CO-stained tissue, both in the flattened tissue, occurred in two steps: coarse, global alignment was via the edges of the asymmetrically-shaped tissue and fine, local alignment was by matching cross-sections of vertical blood vessels in the CO image to putative entry points of ascending/descending vessels in the fluorescence image (*Figure 4F*). Alignment was optimized in anterior V1 and barrel cortex, with the likely result that alignment was most accurate near the visual/somatosensory border. Chemoarchitectonic borders were identified manually.

Cytoarchitectonic borders were examined in 10 Rorb-Ai93 and 11 Emx1-Ai93 mice. Cytoarchitectonic borders were identified from a fluorescence image of the cortical surface via a semi-automated process. Illumination gradients were first removed by filtering the image with a Gaussian kernel (σ = 1290 µm) and subtracting the result. A median filter was used to remove small structures such as blood vessels and edges were detected using Canny edge detection implemented in Python (OpenCV package). From the set of edges, a subset that best matched the borders of primary sensory areas were selected manually. Distances between borders were measured manually, along the axis of the LM/RL border.

## Projection-based retinotopic map

The projection-based map was derived from analysis of data from the Allen Mouse Brain Connectivity Atlas (*Oh et al., 2014*); http://connectivity.brain-map.org/). The Allen Mouse Brain Connectivity Atlas is a large data set derived from many mice, each with a single injection of adenoassociated virus that drives expression of GFP, an anterograde tracer. After fixation, each mouse brain in the atlas was imaged via an automated imaging and sectioning microscope, and registered to a 3-dimensional template derived from the mean autofluorescence of 1675 mouse brains (common coordinate framework v3, http://help.brain-map.org//display/mouseconnectivity/API). Hence the atlas includes the location of each injection site and the distribution of fluorescently-labeled projections from the injection site, with each voxel registered to a three-dimensional template of the mouse brain (*Kuan et al., 2015*).

We selected data from 99 mice (35 wild-type (Bl6) mice and 64 Cre mice), each with an injection into V1. The data set from each mouse consists of a 3D map of projection density. We processed the data sets through several distinct steps (*Figure 6—figure supplement 1*), en route to creating two maps of connectivity anterior-posterior and medial-lateral connectivity which are comparable to maps of altitude and azimuth, respectively. In the first step, we pooled data from 99 mice by

creating a weighted 3D map of connectivity, with the entry in each voxel being the center of mass of all source locations weighted by projection strength:

$$\vec{\mathrm{L}}_{proj} = \frac{\sum_{i=1}^{n} W_i.\vec{\mathrm{L}}_i}{\sum_{i=1}^{n} W_i}$$

where $\vec{\mathrm{L}}_{proj}$ is a vector specifying the estimated source location (in 3D) for this target voxel.

$\vec{\mathrm{L}}$ is a 3D vector specifying the center of an injection site.

$W$ is a measure of projection strength: $W = F_t/F_s$

$F_t$ is the projection density (derived from fluorescence intensity) at the target pixel.

$F_s$ is the injection density (derived from fluorescence intensity) at the source pixels (the injection site).

$n$ = 99 mice

The output of this procedure was a 3D map (a 3D array) in which each voxel contained a vector ($\vec{\mathrm{L}}_{proj}$) with three entries: the x, y and z positions of the location in V1 from which the voxel received the strongest projection.

From this 3D array, we projected $\vec{\mathrm{L}}_{proj}$ to the pial surface of cortex. We first calculated equipotential surfaces, using a Laplace transform, with each point on a surface being equidistant on a normalized scale from pia (distance = 0) to white matter (distance = 1) (http://help.brain-map.org/download/attachments/2818171/MouseCCF.pdf?version=1&modificationDate=1432939552497). At each surface location, the voxel with the greatest summed projection strength along a line orthogonal to the equipotential surfaces was identified and its $\vec{\mathrm{L}}_{proj}$ was projected to the surface, where summed projection strength = $\sum_{i=1}^{n} W_i$. Only $\vec{\mathrm{L}}_{proj}$ values from 0.1 to 0.5 of the normalized cortical depth were projected to the pial surface. The result was a 2D map of $\vec{\mathrm{L}}_{proj}$ entries. This 2D pial surface map was projected to the horizontal plane, yielding a 2D 'top view' of $\vec{\mathrm{L}}_{proj}$ entries.

From the $\vec{\mathrm{L}}_{proj}$ top view, we generated two maps: one displaying the anterior-posterior location in V1 from which each pixel received the strongest projection, and the other displaying the medial-lateral-posterior location in V1 from which each pixel received the strongest projection. As vertical and horizontal retinotopy are represented in orthogonal axes in V1, these two projection maps are comparable to maps of altitude and azimuth. After filtering (σ = 43 µm), we used these two maps to generate the projection-based equivalent of a field sign map, which we term the 'projection sign map' (also filtered, σ = 130 µm).

We processed the projection sign map to locate borders and, thereby, draw a projection-based retinotopic map of visual cortex using the same numerical routine employed to derive borders from field sign maps. The first step in deriving borders is to threshold the sign map. For projection-based signs maps we employed a threshold of ±0.2. Since a threshold of ±0.2 could leave gaps between visual areas, such as at the intersection of V1, AL, RL and LM (e.g. *Figure 6E*), we calculated a second projection-based field sign map using a threshold of ±0.1.

The locations of injection sites and the projection-based retinotopic map were displayed on a surface projection of autofluorescence derived from 1675 mouse brains, thereby revealing the registration of injection sites and the projection-based retinotopic map to the borders of major architectonically-defined cortical areas.

## 2-photon microscopy and comparison with widefield images

2-photon experiments were performed on a Sutter MoM. Before 2-photon imaging, a retinotopic map was generated using widefield GCaMP6 fluorescence, as described above. The mouse was then moved to the 2-photon microscope where a 'local' widefield retinotopic map was generated through the microscope objective using LED illumination, a sCMOS camera and the drifting checkerboard stimulus. The V1-LM border location (which was later compared with single-cell retinotopy derived from 2-photon imaging) was derived from the local widefield map.

To ensure accurate registration of local widefield and 2-photon images, the local widefield map was generated with the microscope objective focused 200 µm below the pial surface of cortex. The

2-photon data set was collected immediately afterwards without axial or transverse translation of the field of view of the microscope, relative to the preparation. The field of view of the widefield image was greater than that of the 2-photon image. For accurate alignment of the two images, images of the surface vasculature were acquired under widefield and 2-photon illumination and used to guide a rigid transform of the widefield image, which was then cropped to the dimensions of the 2-photon image.

2-photon imaging was performed with 920 nm illumination from a Ti:sapphire laser (Coherent Chameleon II), which was focused onto the prep with a x16/0.8 NA objective (Nikon N16XLWD-PF), providing a 720×720 µm field of view. 512×512 pixel images (1.4 µm per pixel) were acquired at 30 Hz. Emitted light was collected in the epifluorescence configuration through a 735 nm dichroic reflector (FF735-DiO1, Semrock) and a 490–560 bandpass emission filter (ET525/70 m-2P, Chroma Technology). Image acquisition was controlled using ScanImage software. Single-cell receptive field mapping was performed using a sparse noise stimulus consisting of black and white squares on a 50% grey background in pseudorandom order. Each square (6×6 visual degrees, 100 ms duration) was displayed 60 times per polarity on an LCD monitor (ASUS PA248Q, mean luminance: 50 cd/m$^2$).

Fluorescence was extracted from the 2-photon time series by defining weighted somatic regions of interest (ROIs) using a PCA-ICA routine (*Mukamel et al., 2009*, µ = 0.2). A size filter was employed to eliminate ROIs smaller than 59 µm$^2$ or larger than 395.5 µm$^2$. For each somatic ROI, a neuropil ROI was created by dilating the outer border of the somatic ROI by 5 and 15 pixels to define inner and outer limits of the neuropil ROI. The union of somatic ROIs was excluded from all neuropil ROIs. For each neuron we extracted two fluorescence values per time point using the two ROIs: $F_{measured}$ from the somatic ROI and $F_{neuropil}$ from the neuropil ROI. We then calculated the true somatic fluorescence (Fc, without neuropil contamination) assuming linear summation: $F_c = F_{measured} + r^*F_{neuropil}$. We estimated r, the contamination ratio, separately for each soma by gradient descendent regression with a smoothness regularization (http://help.brain-map.org/download/attachments/10616846/VisualCoding_Overview.pdf?version=1&modificationDate=1465258498093). The mean ± standard deviation value of r for all neurons in our data set was 0.244 ± 0.189.

After neuropil subtraction, we calculated the stimulus-triggered average fluorescence for each stimulus pixel location and polarity. An example is illustrated in *Figure 7—figure supplement 2A*). For each stimulus location, ΔF/F integrals during the 300 ms following stimulus onset were calculated for white and black squares, yielding On and Off spatial receptive fields respectively. Baseline was defined as mean fluorescence within the 0.5 s window before stimulus onset. From On and Off receptive fields we calculated On and Off z-score maps by subtracting the mean of the pixel values in the map and dividing by the standard deviation of the pixel values in the map. z-score maps were smoothed with a Gaussian filter (σ = 6 degrees) and up-sampled by a factor of 10 with cubic interpolation. If the maximum of either the On or Off receptive field was greater than two, the neuron was considered responsive to the stimulus and was included in subsequent analyses.

To calculate the receptive field center of each cell, for each receptive field map pixels with a z-score below a threshold were set to zero. The threshold employed was 40% of the greater of the maximum On and maximum Off z-scores. Pixels with z-scores above the threshold retained their original z-score values. On and Off thresholded receptive field maps were summed and the altitude and azimuth of the cell's receptive field location were defined as weighted average coordinates of the combined receptive field. The soma-border distance for each soma was the shortest Cartesian distance between the border and the weighted average coordinates of the soma pixels.

## Acknowledgements

Funding was provided by the Allen Institute for Brain Science and award number NS078067 from the National Institute of Neurological Disorders and Stroke. Its contents are solely the responsibility of the authors and do not necessarily represent the official views of the National Institutes of health and the National Institute of Neurological Disorders and Stroke. GCaMP constructs were provided by Janelia Research Campus. We thank the many staff members of the Allen Institute, especially Linda Madisen and Hongkui Zeng for GCaMP6 reporter lines, the In Vivo Sciences team for surgeries, Marc Takeno and Quanxin Wang for advice on histology, and members of the Allen Institute for Brain Science for comments on the manuscript. We thank the Allen Institute founders, Paul G Allen and Jody Allen, for their vision, encouragement and support.

## Additional information

### Funding

| Funder | Grant reference number | Author |
|--------|------------------------|--------|
| Allen Institute for Brain Science | | Jun Zhuang<br>Lydia Ng<br>Derric Williams<br>Yang Li<br>Marina Garrett<br>Jack Waters |
| National Institute of Neurological Disorders and Stroke | NS078067 | Jack Waters |

The funders had no role in study design, data collection and interpretation, or the decision to submit the work for publication.

### Author contributions

JZ, Conception and design, Acquisition of data, Analysis and interpretation of data, Drafting or revising the article; LN, YL, Analysis and interpretation of data, Contributed unpublished essential data or reagents; DW, Contributed unpublished essential data or reagents; MV, Analysis and interpretation of data, Drafting or revising the article; MG, JW, Conception and design, Analysis and interpretation of data, Drafting or revising the article

### Author ORCIDs

Jun Zhuang, http://orcid.org/0000-0002-6237-3270
Jack Waters, http://orcid.org/0000-0002-2312-4183

### Ethics

Animal experimentation: This study was performed in accordance with the recommendations in the Guide for the Care and Use of Laboratory Animals of the National Institutes of Health. All of the animals were handled according to institutional animal care and use committee protocols of the Allen Institute for Brain Science, protocols 1205, 1406 and 1408.

## Additional files

### Supplementary files

• Supplementary file 1. Retinotopic mapping example analysis. Example of derivation of a field sign map from altitude and azimuth maps, in HTML format. The same example is available as a Jupyter notebook at https://github.com/zhuangjun1981/retinotopic_mapping.

### Major datasets

The following previously published dataset was used:

| Author(s) | Year | Dataset title | Dataset URL | Database, license, and accessibility information |
|-----------|------|---------------|-------------|--------------------------------------------------|
| Oh SW, Harris JA, Ng L, Winslow B, Cain N, Mihalas S, Wang Q, Lau C, Kuan L, Henry AM, Mortrud MT, Ouellette B, Nguyen TN, Sorensen SA, Slaughterbeck CR, Wakeman W, Li Y, Feng D, Ho A, Nicholas E, Hirokawa KE, Bohn | 2014 | A mesoscale connectome of the mouse brain | http://connectivity.brain-map.org/ | Publicly available at the Allen Brain Atlas Data Portal |

P, Joines KM, Peng H, Hawrylycz MJ, Phillips JW, Hohmann JG, Wohnoutka P, Gerfen CR, Koch C, Bernard A, Dang C, Jones AR, Zeng H

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
