## [Decision Letter]

Thank you for submitting your article "An extended retinotopic map of mouse cortex" for consideration by *eLife*. Your article has been reviewed by two peer reviewers, and the evaluation has been overseen by David Kleinfeld as Reviewing Editor and Sabine Kastner as the Senior Editor. Tim Murphy (Reviewer #1) has agreed to reveal his identity.

The reviewers have discussed the reviews with one another and the Reviewing Editor has drafted this decision to help you prepare a revised submission.

Overall, the imaging, and you are an expert in this arena, is carefully performed with many technical controls, e.g., autofluorescence, although reviewer 1 has additional requests on this issue. The cytochrome oxidase staining is excellent and the comparisons of this straining, as a measure of the underlying cytoarchitecture, with the functional imaging forms the kernel of the manuscript. The central claim that functional and cytoarchitecural borders do not agree is exhilarating yet at the same time troubling. For that reason, both the reviewers and the Reviewing Editor (BRE) concur that this aspect of the work needs to be both better examined and more fully explained. The BRE has summarized this issue in terms of additional experiments and analysis that should provide clarity. You are not obligated to perform all or even any of these, but you do need to convince the reviewers and BRE that the extension of activation beyond cytological borders is real. Please consider:

1) As you mention in the text, you are recording calcium signals that are very likely to have dendritic contributions that extend outside somatic boundaries. To echo the comments of reviewer 1, it is imperative to know if these extend past cytological boundaries as they do in other areas (e.g., motor nuclei, to bring up classical structures). Perhaps you can do a number of intracellular fills of somata near the calcium signal and see if these, as opposed to the dendrites, are within the cytological border. Or perhaps you can make maps based on the calcium response to more localized cells, like parvalbumen cells, for at least a few areas.

2) Related to the above, the wide field imaging technique will yield a signal that extends beyond a cytological boundary as a result of scattering of the excitation as well as emitted light. Perhaps you could examine the border with two-photon imaging of the Ca-response, which will avoid this issue, in a number of cases. A comparison of calcium signaling as captured by wide-field imaging versus 3-D (z-stacks) with two-photo imaging near a few borders would be an excellent technical contribution to the field and settle many issues regarding this and other studies of this type.

3) Echoing comments of reviewer #2, we are uncomfortable with the analysis based on "visual field sign", which further needs to be clarified early on in the manuscript and not in the Methods. As you note "…visual field sign at each pixel is the sine of the angle between the local gradients… in azimuth and altitude. To find borders, the visual field map was converted to a binary image using a manually-defined threshold (~0.3-0.5) and the initial visual patches were further processed with a split/merge routine (Garrett et al., 2014)…". We assume that the threshold is on the absolute value of the visual field sign, which ranges [-1,+1] before binarization. This critical step needs to be justified with a sensitivity analysis, i.e., would a shift in threshold change the extent of overlap.

4) Related to the above, and again echoing comments of reviewer #2, we are uncomfortable with the smearing of the signal as a result of eye-movement. We appreciate that the SE of the measured mean position is small, but the range of movement is large and this smearing could also appreciably extend the border. Perhaps an extend run should be made on one map and only trials with movement of say < one degree is averaged. This would demonstrate if movement does or does not contribution to a systematic extension of the borders.

Lastly, we come to the issue of the Glasser et al. (Nature 2016 PMID: 27437579) paper. As paraphrased from reviewer 2's summary: "…the mismatch between field maps and cytoarchitectonic maps has not been convincingly demonstrated. This casts doubts on the most far-reaching conclusion that cytoarchitectonic borders cannot be considered as real borders. If true the much celebrated Brodmann 2.0 myeloarchitectonic area map of human cortex (Glasser et al., Nature 2016) should not be considered a map of functionally distinct areas. So, in the absence of conclusive evidence a more cautious conclusion would be that the registration of visual field maps to the underlying anatomy lacks sufficient spatial resolution to definitively settle the issue…". We note that the maps in Glaser are based on cytological boundaries, thickness of myelin (myeloarchitectonic), past knowledge of both function projections, and the results of resting state BOLD fMRI. The BRE was surprised that the resting state data in Glasser et al. gave such similar boundaries as those found from cytology, since resting state BOLD is based on parcelating voxels that have similar ultra-low-frequency fluctuations in the BOLD signal (e.g., Smith et al. TiCS 2013). Your method for parcellation is based directly on neuronal (albeit Ca and not spiking) and yields a different result than the resting state BOLD. Thus your conclusions, properly supported, could have profound impact on our thinking of maps derived from resting state BOLD signals versus more direct measures of neuronal activity.

We also ask that you address each of the additional points raised by reviewers. Please note that we expect to receive a revision within two months time.

*Reviewer #1:*

By using a new generation of transgenic mice expressing highly sensitive indicators of neuronal activity and combining with multiple anatomical techniques, this study extends our knowledge of the functional organization of mouse visual cortex. This work clearly follows previous important contributions of Marshel et al. 2011 in Neuron or Garrett et al. 2014 in J. Neurosci which revealed the detailed retinotopic mapping of areas beyond primary visual cortex in mice.

The advantage/novelty of this new work is faster mapping and the identification of additional extra-striate areas and their relationship to other sensory modalities. The authors also effectively use neuronal subset lines for mesoscale mapping. Here, up to 15 putative extrastriate areas have been mapped, some of them, overlapping with territories of other modalities (e.g. barrel cortex). The overlap with other sensory modalities is a new exciting finding. The techniques used here were particularly well implemented and an appropriate amount of controls have been used to validate these findings. Beyond the important descriptive aspect of this work and the high quality of the results, the conclusions open new insight about multimodal processing operating within cortex.

The first part of the study about the mouse line validation appears out of focus and may weaken the impact. To my opinion, this paper is not a technical report to characterize the lines and Figure 1 and Figure 2 could be removed or put in supplementary material. There is no clear conclusion about which mice are best.

Important:

The limitation of the mesoscopic imaging on the topographic and territory boundary mapping are not discussed. What could be the influence of the spread (optical or neuropil)? Can the method reveal visually driven areas not retinotopically organized (i.e. can we evoke the same territories with full field gratings)? To validate the method, the authors should provide the consistency of retinotopic mapping within single animals between days so we can see whether these are truly relatively invariant maps or whether reflect the nuances of the statistical analysis. Is there any consistent relationship with the map borders and surface vessels, maybe an example could be shown?

Figure 7 uses anatomical projection data to predict the sign map. However, the results and methods related to Figure 7 are poorly described. The author should present individual tracing data and show how this is used to generate the projection-based maps. This seems to be a very interesting approach but we cannot understand how you go from anatomical projections to the sign map, more explanation and intermediate steps are shown. NB: the URL in the second paragraph of the subsection “Projection-based retinotopic map” doesn't work. Can you also provide interpretation about the absence of projection to M2/AC recently shown by Murakami et al. 2015 in Front. Mol. Neurosci.

Some information about the origin of the signal could be clarified better: subsection “GCaMP6 expression patterns and fluorescence signals”, third paragraph – “decline in fluorescence that likely results from vasodilation". Do you have any proof this could not be a neuronal suppression? In the fourth paragraph of the aforementioned subsection: “presumably because our fluorescence measurements were from large populations of neurons, which likely display a distribution of spike times" and “These kinetics are unlikely to offer the necessary temporal resolution to reliably follow the flow of visually-evoked activity through visual cortex, which occurs on a timescale of tens of milliseconds". Can you provide the information about the neuronal dynamic using 2-photon on similar mice? Describe how fast is it and how mesoscopic imaging could be considered as a simple additive model?

Figure 5 is beautiful example of how to do correlative anatomy and mesoscale imaging. It would be important to show vessels (arterioles) that are marked on the in vivo image in 5A and follow them through to the flattened and the anatomical map. Some confusion arises since not vessels are visible in the fixed prep (arterioles only?). 2-photon cellular imaging may be another way of confirming that cellular retinotopically mapped signals are present in barrel cortex. However, we recognize that it may be difficult to get the mesoscale field of view using 2P.

*Reviewer #2:*

The paper describes an elegant, fast and deceptively straightforward calcium imaging method for delineating retinotopically organized areas in mouse visual cortex. The results suggest a more complex parcellation scheme with 16 rather than 10 distinct areas. Three of the newly discovered visual areas reside in barrel cortex, encroach onto auditory cortex and take up space in retrosplenial cortex. The results further suggest that retinotopic and cytoarchitectonic borders are misaligned and are therefore ill-suited for delineating cortical areas. Each of the observations is eye-popping and challenges established concepts of cortical parcellation. Although field sign mapping is a powerful tool to provide a rough estimate of visual areas, I am concerned that the method overstates the spatial precision with which this can be achieved. The authors have made an impressive effort to align retinotopic maps with cytoarchitectonically defined areas, but despite the sweat have not crossed the threshold for disposing the long-held notion that cytoarchitectonic borders coincide with retinotopic borders. Three reasons may account for this. First, the eye movements account for up to 375 μm error along azimuth and ~300 μm along elevation, sufficient to account for the suggested misalignment of borders. Second, in the critical Figure 5 it is difficult to match the blood vessels seen during retinotopic mapping with those seen after staining fixed sections for cytochrome oxidase. Further, it is not clear whether these procedures have been applied in all cases. None of the illustrated examples match the maps shown in A, making it difficult to accept the conclusion that borders are mismatched. Third, the patches identified in S1 and retrosplenial cortex map tiny regions of the visual field and resemble more projection fields than separate visual area. Visual projections to both of these regions are well documented. Thus, it appears premature to view RRL and retrosplenial cortex as new visual areas. In summary, the paper makes bold assertions, which if proven correct, have profound consequences.

Subsection “GCaMP6 expression patterns and fluorescence signals”: “GCaMP6 appears not to be evenly distributed across layers”. The statement is in conflict with Figure 1. The distribution of GCaMP6 across extrastriate visual cortex is not clearly shown.

Subsection “Retinotopic maps from GCaMP6 fluorescence reveal additional patches of retinotopic organization”, first paragraph: Provide a detailed description how the "automated routine" identified reliably areal borders. Estimate the precision with which this was done.

Figure.4: Why is RRL only present inEmx-Ai96 mice? Why is RL so much bigger in Emx-Ai96 than other lines? Why is POR absent in all the images? What is "M"? It seems that the "mean map" (Figure 4) is heavily weighted toward Emx-Ai96 and reflects a pattern that depends on the mouse line used for mapping.

Subsection “Retinotopic maps from GCaMP6 fluorescence reveal additional patches of retinotopic organization”, fourth-sixth paragraphs: The usage of "area", "region" and "patch" for describing activated parcel is confusing and blurs the definition of an area.

Figure 5: I am unable to see the overlap in the vascular patterns and therefore cannot match the retinotopic map to the cytochrome oxidase pattern.

Figure 6: What is the evidence that *Rorb* is restricted to V1?

Figure 7: The tracing "experiment" lacks the precision required to conclusively demonstrate that cytoarchitectonic and retinotopic maps are misaligned.

Subsection “Extension of retinotopic organization”, first paragraph: As proposed, area P extends all around the posterior margin of V1. The authors admit that this may be a vascular artifact. Why then is this not acknowledged in Figure 9?

Subsection “Representation of visual space and the organization of mouse visual areas”, last paragraph: It is true that none of the higher visual areas were shown to contain a complete visual hemifield representation. But I found that the reported maps are far more complete than those shown in Figure 8. Thus, the statement is misleading.

---

## [Author Response]

*Overall, the imaging, and you are an expert in this arena, is carefully performed with many technical controls, e.g., autofluorescence, although reviewer 1 has additional requests on this issue. The cytochrome oxidase staining is excellent and the comparisons of this straining, as a measure of the underlying cytoarchitecture, with the functional imaging forms the kernel of the manuscript. The central claim that functional and cytoarchitecural borders do not agree is exhilarating yet at the same time troubling. For that reason, both the reviewers and BRE concur that this aspect of the work needs to be both better examined and more fully explained. The BRE has summarized this issue in terms of additional experiments and analysis that should provide clarity. You are not obligated to perform all or even any of these, but you do need to convince the reviewers and BRE that the extension of activation beyond cytological borders is real. Please consider:*

*1) As you mention in the text, you are recording calcium signals that are very likely to have dendritic contributions that extend outside somatic boundaries. To echo the comments of reviewer 1, it is imperative to know if these extend past cytological boundaries as they do in other areas (e.g., motor nuclei, to bring up classical structures). Perhaps you can do a number of intracellular fills of somata near the calcium signal and see if these, as opposed to the dendrites, are within the cytological border. Or perhaps you can make maps based on the calcium response to more localized cells, like parvalbumen cells, for at least a few areas.*

We agree that the likely extension of dendritic signals across boundaries is an important question. We attempted to address this question using Cre lines that drive sparse GCaMP6 labeling. We have been using primarily Cux2-CreERT2 for these experiments, principally because density of labeled neurons is controllable in Cux2-CreERT2. Unfortunately the minimum labeling density has proven too dense. We may return to this question in the near future since we have a mouse line under construction with the specific aim of controlling labeling density over a greater range. In the meantime, we are unable to answer the question of whether dendritic calcium signals extend across borders, but our widefield-2P comparison indicates that if there is extension of dendritic calcium signals across boundaries then this extension does not prevent us from accurately locating boundaries using widefield fluorescence imaging.

We have added a new section to the Discussion in which we address signals from dendrites that cross borders ('Limitations of mapping with population imaging and simple visual stimuli').

*2) Related to the above, the wide field imaging technique will yield a signal that extends beyond a cytological boundary as a result of scattering of the excitation as well as emitted light. Perhaps you could examine the border with two-photon imaging of the Ca-response, which will avoid this issue, in a number of cases. A comparison of calcium signaling as captured by wide-field imaging versus 3-D (z-stacks) with two-photo imaging near a few borders would be an excellent technical contribution to the field and settle many issues regarding this and other studies of this type.*

We obtained 2-photon data sets aligned to our widefield maps, focusing on the V1-LM border since this is where we have measured the mismatch in architectonic and retinotopic borders. The comparison confirms that widefield borders accurately report the reversal in somatic retinotopy, with a precision of ~30 µm. This precision is more than adequate to support the measured mismatch between architectonic and retinotopic borders of 100-300 µm.

The comparison is presented in a new section of the Results (‘Single-cell retinotopy along the V1-LM border’), with a new Figure 7 and two supplementary figures (Figure 7—figure supplement 1 and Figure 7—figure supplement 2) and we have added 2-photon results to the Github repository.

*3) Echoing comments of reviewer #2, we are uncomfortable with the analysis based on "visual field sign", which further needs to be clarified early on in the manuscript and not in the Methods. As you note "…visual field sign at each pixel is the sine of the angle between the local gradients… in azimuth and altitude. To find borders, the visual field map was converted to a binary image using a manually-defined threshold (~0.3-0.5) and the initial visual patches were further processed with a split/merge routine (Garrett et al., 2014)…". We assume that the threshold is on the absolute value of the visual field sign, which ranges [-1,+1] before binarization. This critical step needs to be justified with a sensitivity analysis, i.e., would a shift in threshold change the extent of overlap.*

The threshold is on the absolute value of the visual field sign. We performed sensitivity analysis to explore the effect of threshold on patch incidence, shape and border location. We compared segmentation of the mean sign map (Figure 3) with thresholds of 0.2, 0.3 and 0.4. Thresholds of 0.2 and 0.4 are the limits of the range of values employed in the analysis of our data sets and a threshold of 0.3 is close to the mean value (mean = 0.32). These results are presented in a new figure (Figure 2—figure supplement 2) and indicate that:

1) Borders between two patches are generally stable unless the threshold is changed enough to eliminate the border. For example, all the borders of V1 are virtually invariant across the threshold range from 0.2 to 0.4; borders between RL and LM and between LM and P are unaffected by the change in threshold from 0.2 to 0.3, but are absent at a threshold of 0.4; the borders between RL and LLA and between AL and LI are detected by at a threshold of 0.3 or 0.4, but not 0.2. Stable borders are readily identified in the images in panel B, in which persistent borders are in white.

2) Borders that mark the exterior of the map retract as the threshold is raised. This effect is most readily observed through the images in panel C.

3) Many patches persist across the range of thresholds. The patches that are lost at the extreme threshold values are merged with neighboring patches. At a threshold of 0.2, previously published areas that are lost by merger include RL and LLA, and AL and LI. At a threshold of 0.4, previously published areas that are lost by merger include RL, LM and P. One might therefore regard 0.2 and 0.4 as beyond the range of appropriate threshold values for our data sets.

4) For most patches, changing the threshold has little effect on coverage (panel D): the change in coverage is less than the difference in coverage between patches (until the threshold is raised sufficiently to eliminate the patch). As expected, the change in coverage is more pronounced for patches on the periphery of the map that display weaker changes in fluorescence to the visual stimulus, such as patches LLA and RLL.

The effects of changing the sign map threshold, including the effects on visual coverage, are addressed in the second paragraph of the subsection “Retinotopic maps from GCaMP6 fluorescence reveal additional patches of retinotopic organization” of the Results and in the legend to Figure 2—figure supplement 2.

We have also made changes to the manuscript to further explain our analysis routines:

1) In the Methods, we expanded the description of the processing steps from sign map to borders, including more information on the thresholding step and values of the variables employed (subsection “Widefield image analysis”, third paragraph).

2) We have added a supplementary figure (Figure 2—figure supplement 1), illustrating with an example the analysis workflow from sign map to borders.

3) To the Methods, we have added further information about the analysis code placed on Github (subsection “Widefield image analysis”, fourth paragraph).

4) Within the analysis code on Github, we have added descriptions of all variables and information on the values we used when processing our data sets.

5) To the Results, we have added a brief description of the analysis routine (subsection “Retinotopic maps from GCaMP6 fluorescence reveal additional patches of retinotopic organization”, second paragraph).

*4) Related to the above, and again echoing comments of reviewer #2, we are uncomfortable with the smearing of the signal as a result of eye-movement. We appreciate that the SE of the measured mean position is small, but the range of movement is large and this smearing could also appreciably extend the border. Perhaps an extend run should be made on one map and only trials with movement of say < one degree is averaged. This would demonstrate if movement does or does not contribution to a systematic extension of the borders.*

We added a supplementary figure (Figure 2—figure supplement 4) in which we compared maps in a single imaging session from an Ai96 mouse, sorting trials into those with and without eye movements of greater than 2 degrees. The exterior borders of the sign map displayed some differences between maps, either as a result of eye movements or of averaging across the relatively small number of trials without eye movements. Importantly, the borders between visual areas (including that between V1 and LM/RL) were largely unaffected by eye movements, leading us to conclude that eye movements are not responsible for the mismatch in border locations that we report.

*Lastly, we come to the issue of the Glasser et al. (Nature 2016 PMID: 27437579) paper. As paraphrased from reviewer 2's summary: "…the mismatch between field maps and cytoarchitectonic maps has not been convincingly demonstrated. This casts doubts on the most far-reaching conclusion that cytoarchitectonic borders cannot be considered as real borders. If true the much celebrated Brodmann 2.0 myeloarchitectonic area map of human cortex (Glasser et al., Nature 2016) should not be considered a map of functionally distinct areas. So, in the absence of conclusive evidence a more cautious conclusion would be that the registration of visual field maps to the underlying anatomy lacks sufficient spatial resolution to definitively settle the issue…". We note that the maps in Glaser are based on cytological boundaries, thickness of myelin (myeloarchitectonic), past knowledge of both function projections, and the results of resting state BOLD fMRI. The BRE was surprised that the resting state data in Glasser et al. gave such similar boundaries as those found from cytology, since resting state BOLD is based on parcelating voxels that have similar ultra-low-frequency fluctuations in the BOLD signal (e.g., Smith et al. TiCS 2013). Your method for parcellation is based directly on neuronal (albeit Ca and not spiking) and yields a different result than the resting state BOLD. Thus your conclusions, properly supported, could have profound impact on our thinking of maps derived from resting state BOLD signals versus more direct measures of neuronal activity.*

Our new 2P data set indicates that our widefield imaging results offer sufficient precision to measure the mismatch in cytoarchitectonic and functional borders. However, the presence of a mismatch does not mean that 'cytoarchitectonic borders cannot be considered as real borders*'*. Architectonic borders do not necessarily match retinotopic borders, but may align with other functional borders. For example, it is possible that the retinotopic lateral border of V1 marks the lateral extent of retino-geniculate projections from the contralateral eye representing the contralateral visual field, and the architectonic border the lateral extent of retino-geniculate projections from the contralateral eye representing the ipsilateral visual field (Laing et al., 2015).

With regard to Glaser et al., who report on areas in humans, we would advocate caution when generalizing across species. Although there is evidence of a mismatch in structural and functional border locations for some higher visual areas in humans (Large et al., 2016, Cerebral Cortex 26, 3928–3944), we do not know whether there is a mismatch in the V1-V2 border locations in humans, which might be the equivalent of the V1-LM/RL border mismatch we have observed. If a V1-V2 border mismatch exists in humans, the size of the mismatch will need to be determined. Possibly it may be too small to be determined with fMRI-based functional measurements.

*We also ask that you address each of the additional points raised by reviewers. Please note that we expect to receive a revision within two months time.*

*Reviewer #1:*

*[…] The first part of the study about the mouse line validation appears out of focus and may weaken the impact. To my opinion, this paper is not a technical report to characterize the lines and Figure 1 and Figure 2 could be removed or put in supplementary material. There is no clear conclusion about which mice are best.*

We have moved Figure 1 into supplementary material (as Figure 1—figure supplement 1). Figure 2, describing the amplitude and kinetics of the GCaMP6 impulse-response in our mice, remains in the main body of the paper (as Figure 1), but we condensed the text associated with this figure.

*Important:*

*The limitation of the mesoscopic imaging on the topographic and territory boundary mapping are not discussed. What could be the influence of the spread (optical or neuropil)? Can the method reveal visually driven areas not retinotopically organized (i.e. can we evoke the same territories with full field gratings)?*

We have added a section to the Discussion in which we address limitations of mesoscopic imaging, including optical and neuropil spread (subsection “Limitations of mapping with population imaging and simple visual stimuli”).

Regarding full field gratings and the responsiveness of regions that are not retinotopically-organized, visually-driven regions can be extracted from power maps after pixel-wise FFT analysis (Figure 2, panel B). However, one would need to threshold the power map to define border locations. The power map is not sensitive to retinotopy, but aligns well to the sign map, leading us to conclude that visual responsiveness and retinotopy tend to correlate. However, the relationship appears weaker in the medial part of the window, where retrosplenial cortex responded to visual stimuli but displayed relatively weak (or noisy) retinotopy (Figure 3 panel A, B, C). In our preparation, optical access to medial cortex is limited due to the placement of the window and the signal-to-noise ratio in this part of the window was probably poorer than in more lateral locations. Further studies, in a modified preparation, would probably be necessary to more fully understand the relationship between the amplitude of visually-evoked responses and retinotopy in medial cortex, although we should note that in a preparation with a more medial window location, the central sinus might limit optical access.

*To validate the method, the authors should provide the consistency of retinotopic mapping within single animals between days so we can see whether these are truly relatively invariant maps or whether reflect the nuances of the statistical analysis. Is there any consistent relationship with the map borders and surface vessels, maybe an example could be shown?*

We have not performed an extensive or quantitative study of the consistency of maps across sessions, but maps generally appear fairly stable. Stability is typically greater for areas and borders near the center of the map, as we might expect given that the amplitude of the visually-evoked change in fluorescence is greater towards the center of the map (Figure 2).

We have added a new figure in which we display examples of repeated imaging for two Emx1-Ai96 mice, with imaging sessions separated by almost 90 days (Figure 3—figure supplement 1; subsection “Retinotopic maps from GCaMP6 fluorescence reveal additional patches of retinotopic organization”, fifth paragraph). Even over this long time period, maps acquired from the same mouse appear more similar than maps from different mice (panels A and B). For each mouse, surface vasculature is stable and border positions shift little relative to the surface vasculature (panels C and D). As a simple quantification of variability, we plot patch areas (panels E and F). As expected, areas are fairly stable between sessions with variability being greatest for the small, peripheral patches.

*Figure 7 uses anatomical projection data to predict the sign map. However, the results and methods related to Figure 7 are poorly described. The author should present individual tracing data and show how this is used to generate the projection-based maps. This seems to be a very interesting approach but we cannot understand how you go from anatomical projections to the sign map, more explanation and intermediate steps are shown.*

We have added more information to the Methods section to further explain the processing of projection data sets (subsection “Projection-based retinotopic map”, second to fifth paragraphs). In addition, we have created Figure 6—figure supplement 1 that lists the intermediate steps, in order, and provides a simple graphical representation intended to illustrate some key steps.

*NB: the URL in the second paragraph of the subsection “Projection-based retinotopic map” doesn't work.*

We tested the URL on several machines, with Thunderbird, Chrome and Explorer, using the hyperlink and the URL typed manually. In all cases we were directed to the correct GitHub page. When manually entering the URL, we initially typed the wrong URL because the URL was underlined, which obscured an underscore between 'retinotopic' and 'mapping': https://github.com/zhuangjun1981/retinotopic_mapping

To avoid this problem, in the revised manuscript (subsection “Widefield image analysis”, fourth paragraph) the URL is not underlined, revealing the underscore: https://github.com/zhuangjun1981/retinotopic_mapping.

*Can you also provide interpretation about the absence of projection to M2/AC recently shown by Murakami et al. 2015 in Front. Mol. Neurosci.*

M2/AC, as identified by Murakami et al. (2015) is likely outside the cranial window in our experiments. Our window was circular and centered over posterior cortex, 2.7 mm lateral to the midline. Posterior S1 was typically along the anterior edge of the window. M2/AC is slightly anterior to posterior S1 and almost on the midline, with the result that M2/AC is anterior and medial to our window. To further illustrate the relative positions of these regions and our cranial window, Figure 10 illustrates the approximate position of our 5 mm diameter cranial window (dashed line, to scale). Image taken from Figure 2 of Murakami et al. (2015).

Author response image 1.**DOI:**
http://dx.doi.org/10.7554/eLife.18372.028

Some information about the origin of the signal could be clarified better: subsection “GCaMP6 expression patterns and fluorescence signals”, third paragraph – “decline in fluorescence that likely results from vasodilation". Do you have any proof this could not be a neuronal suppression?

We do not have proof. We have moderated this statement to indicate that vasodilation is one possible explanation for the decline in fluorescence. The text now reads '…that may result from vasodilation'.

*In the fourth paragraph of the aforementioned subsection: “presumably because our fluorescence measurements were from large populations of neurons, which likely display a distribution of spike times" and “These kinetics are unlikely to offer the necessary temporal resolution to reliably follow the flow of visually-evoked activity through visual cortex, which occurs on a timescale of tens of milliseconds". Can you provide the information about the neuronal dynamic using 2-photon on similar mice? Describe how fast is it and how mesoscopic imaging could be considered as a simple additive model?*

In the interests of condensing the text associated with the first two figures, we have deleted these sections of text.

*Figure 5 is beautiful example of how to do correlative anatomy and mesoscale imaging. It would be important to show vessels (arterioles) that are marked on the in vivo image in 5A and follow them through to the flattened and the anatomical map. Some confusion arises since not vessels are visible in the fixed prep (arterioles only?). 2-photon cellular imaging may be another way of confirming that cellular retinotopically mapped signals are present in barrel cortex. However, we recognize that it may be difficult to get the mesoscale field of view using 2P.*

The fluorescently-tagged lectin we employed apparently labels only a subset of vessels. We do not know which subset (possibly arterioles), but the labeling is extensive enough to permit accurate alignment of images. To help readers match vessels between images, we have added a supplementary figure in which we have traced some of the labeled vessels that persist from the in vivo image through the perfused, flattened and cytochrome oxidase images (Figure 4—figure supplement 1).

*Reviewer #2:*

*The paper describes an elegant, fast and deceptively straightforward calcium imaging method for delineating retinotopically organized areas in mouse visual cortex. The results suggest a more complex parcellation scheme with 16 rather than 10 distinct areas. Three of the newly discovered visual areas reside in barrel cortex, encroach onto auditory cortex and take up space in retrosplenial cortex. The results further suggest that retinotopic and cytoarchitectonic borders are misaligned and are therefore ill-suited for delineating cortical areas. Each of the observations is eye-popping and challenges established concepts of cortical parcellation. Although field sign mapping is a powerful tool to provide a rough estimate of visual areas, I am concerned that the method overstates the spatial precision with which this can be achieved. The authors have made an impressive effort to align retinotopic maps with cytoarchitectonically defined areas, but despite the sweat have not crossed the threshold for disposing the long-held notion that cytoarchitectonic borders coincide with retinotopic borders. Three reasons may account for this. First, the eye movements account for up to 375 μm error along azimuth and ~300 μm along elevation, sufficient to account for the suggested misalignment of borders.*

As discussed above (under '(4) Related to the above, and again echoing comments of reviewer #2'), we have addressed eye movements in a new Figure 3—figure supplement 3, concluding that eye movements had little effect on the alignment of borders between visual areas.

*Second, in the critical Figure 5 it is difficult to match the blood vessels seen during retinotopic mapping with those seen after staining fixed sections for cytochrome oxidase. Further, it is not clear whether these procedures have been applied in all cases. None of the illustrated examples match the maps shown in A, making it difficult to accept the conclusion that borders are mismatched.*

We have made a supplementary figure (Figure 4—figure supplement 1) that highlights some of the vessels that are visible across the sequence of experimental conditions. The same analysis procedures, described in the Methods, were applied in every case. In Figure 4 (previously Figure 5), panels A to G are from the same mouse. The sign maps in panels A and G are at different scales (scale bars are in panels D and G) and the in vivo images in panel A need to be warped slightly (as described in the methods) to accurately match the images after fixation. To illustrate the match, in Figure 11 we have scaled and overlaid sign maps from panels A and G of Figure 4 (without the warping that's necessary for precise alignment).

Author response image 2.**DOI:**
http://dx.doi.org/10.7554/eLife.18372.029

*Third, the patches identified in S1 and retrosplenial cortex map tiny regions of the visual field and resemble more projection fields than separate visual area. Visual projections to both of these regions are well documented. Thus, it appears premature to view RRL and retrosplenial cortex as new visual areas. In summary, the paper makes bold assertions, which if proven correct, have profound consequences.*

We agree with the reviewer. Our results indicate that there is retinotopic organization in RLL and retrosplenial cortex, but do not indicate whether somata in these regions are retinotopically organized. We deliberately avoid applying the term 'visual area' to RLL and to retrosplenial cortex, instead referring to them as field sign 'patches' and as regions with retinotopic organization. We have expanded on the discussion of this topic to further clarify our use of the terms 'area', 'patch' and 'region' (subsection “Cortical regions, field sign patches and visual areas”, last paragraph).

*Subsection “GCaMP6 expression patterns and fluorescence signals”: “GCaMP6 appears not to be evenly distributed across layers”. The statement is in conflict with Figure 1. The distribution of GCaMP6 across extrastriate visual cortex is not clearly shown.*

Fluorescence is evenly distributed across layers in Emx1-Ai95 mice. The distribution is less clear for Emx1-Ai96, in which expression is weaker, but both show relatively homogenous fluorescence across layers compared to Emx1-Ai93. To avoid confusion, we have re-written this sentence, which is now in the legend for Figure 1—figure supplement 1.

We have not analyzed the laminar distribution across visual areas, beyond determining that GCaMP6 is expressed throughout cortex. In part this is because the borders between areas are difficult to locate in fixed sections, but also because our results (expressed as △F/F changes in fluorescence) should be relatively insensitive to differences in expression of GCaMP6 across areas. Hence our expression data tell us what we need to know for the purposes of our study: that GCaMP6 is expressed throughout cortex in all three Emx1-GCaMP6 lines.

*Subsection “Retinotopic maps from GCaMP6 fluorescence reveal additional patches of retinotopic organization”, first paragraph: Provide a detailed description how the "automated routine" identified reliably areal borders. Estimate the precision with which this was done.*

We have added further explanation of the automated routine to the manuscript and performed sensitivity analysis on the sign map thresholding step, described above in our responses to the consolidated comments (under '(3) Echoing comments of reviewer #2…'). To address precision, we compare widefield borders to single-cell tuning using 2-photon calcium imaging. Again, these experiments are described in our responses to the consolidated comments (under '(2) Related to the above…').

*Figure.4: Why is RRL only present inEmx-Ai96 mice?*

RLL was present in 3 of 10 (30%) Emx1-Ai93 mice and 3 of 4 (75%) Emx1-Ai96 mice. We have added these numbers to the Results (subsection “Retinotopic maps from GCaMP6 fluorescence reveal additional patches of retinotopic organization”, tenth paragraph). We're reluctant to speculate on whether this difference in incidence of RLL indicates a biological difference between mouse lines. RLL exhibits the smallest amplitude change in fluorescence of any field sign patch (Figure 3). We would speculate that determining whether RLL occurs preferentially in one of other mouse line would require maps from many more mice. Even with additional maps, it may be difficult to distinguish between a difference incidence of RLL between mouse lines and an apparent difference in incidence due to differing signal-to-noise between lines.

*Why is RL so much bigger in Emx-Ai96 than other lines?*

Comparing Emx1-Ai93 and Emx1-Ai96, there is a difference in the size of some patches, including RL (Figure 12). We don't yet know the reason(s) for these differences. We're cautious about interpreting these results since, once split by mouse line, each group contains data from small numbers of mice. Nonetheless, some of these differences are statistically significant. We suspect the differences in area may be related to the laminar expression patterns of GCaMP6. We're looking into this issue further, are currently gathering maps from several Cre lines and plan to address the topic in a future manuscript. Importantly, however, the main conclusions of our manuscript hold across mice and mouse lines, with the size and border locations V1, for example, being invariant.

Author response image 3.**DOI:**
http://dx.doi.org/10.7554/eLife.18372.030

*Why is POR absent in all the images?*

POR was likely outside the cranial window in most, perhaps all of our experiments (subsection “Retinotopic maps from GCaMP6 fluorescence reveal additional patches of retinotopic organization”, ninth paragraph).

*What is "M"?*

M or 'medial region' is a negative field sign patch located medial to V1 and posterior to PM (Garrett et al., 2014).

*It seems that the "mean map" (Figure 4) is heavily weighted toward Emx-Ai96 and reflects a pattern that depends on the mouse line used for mapping.*

The mean map (now Figure 3) is the mean of the Emx1-Ai93 and Emx1-Ai96 maps in panel B. We combined these two maps with equal weighting to avoid biasing the mean map towards Emx1-Ai93 mice. (The data set contains maps from 10 Emx1-Ai93 mice and 4 Emx1-Ai96 mice).

*Subsection “Retinotopic maps from GCaMP6 fluorescence reveal additional patches of retinotopic organization”, fourth-sixth paragraphs: The usage of "area", "region" and "patch" for describing activated parcel is confusing and blurs the definition of an area.*

We have used 'area', 'region' and 'patch' deliberately throughout the document, with each having a different meaning. Unfortunately, we had not stated explicitly the meaning of each term, as we had used them. We have added a section to the Discussion (entitled 'Cortical regions, field sign patches and visual areas') in which we address the topic of somatic vs. axonal retinotopy and explicitly state how we have used each of these terms (last paragraph).

*Figure 5: I am unable to see the overlap in the vascular patterns and therefore cannot match the retinotopic map to the cytochrome oxidase pattern.*

We have made a supplementary figure (Figure 4—figure supplement 1) that highlights some of the vessels that are visible across the sequence of experimental conditions.

*Figure 6: What is the evidence that Rorb is restricted to V1?*

We have addressed the distribution of fluorescence in *Rorb*-IRES-Cre mice in our responses to reviewer #1, above.

*Figure 7: The tracing "experiment" lacks the precision required to conclusively demonstrate that cytoarchitectonic and retinotopic maps are misaligned.*

The tracing 'experiment' is one of several methods we employed to compare retinotopic and architectonic border locations. We are not able to quantify the precision of these methods with confidence, but we believe that all are likely to estimate the architectonic border location with sufficient precision to support the conclusions of the manuscript, that borders are mismatched by hundreds of micrometers. Our confidence in these methods is supported by the fact that they each provide a similar estimate of the mismatch.

For a more precise measure of the precision with which we can locate borders, we have now used single-cell retinotopy, adding a 2-photon data set that indicates our widefield border locations are likely accurate to within a few tens of micrometers.

*Subsection “Extension of retinotopic organization”, first paragraph: As proposed, area P extends all around the posterior margin of V1. The authors admit that this may be a vascular artifact. Why then is this not acknowledged in Figure 9?*

We do not believe that the extension of P across the posterior extent of V1 is a vascular artifact since there are projections from V1 into this region of cortex. Functional evidence for retinotopic organization posterior to V1 is more sparse, but Garrett et al. (2014) includes reflectance images in which there appears to be retinotopic organization posterior to V1 (Garrett et al., Figure 2). We can only speculate on why these posterior regions have not mapped more consistently in the past, but the additional challenges of imaging in this posterior location (the posterior sinus, the folded structure of posterior cortex and, in our experiments, the edge of the cranial window) may reduce the signal-to-noise ratio, relative to other parts of visual cortex, leading to less consistent results. In our maps, there is consistent retinotopic structure posterior to V1, with a field sign-positive strip extending across the posterior extent of V1 in almost every mouse (Figure 3). Possibly the high signal-to-noise ratio of GCaMP6 imaging (compared to the SNR of reflectance-based imaging employed by many previous authors) may be sufficient to resolve consistent structure posterior to V1.

We consider this topic, including comparison with the literature, more appropriate for the Discussion than the Results section, hence its exclusion from the text associated with Figure 9 and inclusion in the first paragraph of the Discussion subsection “Extension of retinotopic organization”.

*Subsection “Representation of visual space and the organization of mouse visual areas”, last paragraph: It is true that none of the higher visual areas were shown to contain a complete visual hemifield representation. But I found that the reported maps are far more complete than those shown in Figure 8. Thus, the statement is misleading.*

The statement is:

“Even allowing for some expansion of coverage for each visual area, our results indicate that no higher visual areas (even those immediately surrounding V1) contain a complete description of the visual hemifield.”

We were careful to limit this statement to our results to ensure that it was accurate. The new Figure 8—figure supplement 1, in which we illustrate the effects of expanding coverage by the receptive field radius, further supports the statement. We have difficulty in seeing why this statement might be considered misleading and have therefore not changed the statement in the revised manuscript, but would of course be happy to modify the statement if there's a misleading implication that we have overlooked.